

# Supersymmetric ground states of 3D $\mathcal{N} = 4$ gauge theories on a Riemann surface

**Mathew Bullimore[1], Andrea Ferrari[1,2] and Heeyeon Kim[3]**

**1** Department of Mathematical Sciences, Durham University, Lower Mountjoy ,
Stockton Road, Durham, DH1 3LE, UK
**2** Mathematical Institute, University of Oxford, Woodstock Road, Oxford, OX2 6GG, UK
**3** NHETC and Department of Physics and Astronomy, Rutgers University ,
126 Frelinghuysen Rd., Piscataway NJ 08855, USA

## Abstract

This paper studies supersymmetric ground states of 3d $\mathcal{N} = 4$ supersymmetric gauge theories on a Riemann surface of genus $g$. There are two distinct spaces of supersymmetric ground states arising from the $A$ and $B$ type twists on the Riemann surface, which lead to effective supersymmetric quantum mechanics with four supercharges and supermultiplets of type $\mathcal{N} = (2,2)$ and $\mathcal{N} = (0,4)$ respectively. We compute the space of supersymmetric ground states in each case, graded by flavour and R-symmetries and in different chambers for real mass and FI parameters, for a large class of supersymmetric gauge theories. The results are formulated geometrically in terms of the Higgs branch geometry. We perform extensive checks of compatibility with the twisted index and mirror symmetry.



# 1 Introduction

This paper studies the supersymmetric ground states of 3d $\mathcal{N} = 4$ gauge theories on $\mathbb{R} \times \Sigma$, where $\Sigma$ is a Riemann surface of genus $g$. There are two distinct spaces of supersymmetric ground states depending on which R-symmetry is chosen to twist along the Riemann surface $\Sigma$. They are referred to as the *A*-twist and *B*-twist and are exchanged by three-dimensional mirror symmetry.

The strategy, following a similar philosophy in [1, 2], is to introduce an effective supersymmetric quantum mechanics on $\mathbb{R}$ that captures the supersymmetric ground states of the system. The type of supersymmetric quantum mechanics depends on the twist:

- The *A*-twisted supersymmetric ground states are captured by an *A*-type $\mathcal{N} = 4$ quantum

mechanics, with supermultiplets obtained by dimensional reduction of 2d $\mathcal{N} = (2,2)$ vectormultiplets and chiral multiplets.

- The $B$-twisted supersymmetric ground states are captured by a $B$-type $\mathcal{N} = 4$ quantum mechanics, with supermultiplets obtained by dimensional reduction of 2d $\mathcal{N} = (0,4)$ vectormultiplets, hypermultiplet and twisted hypermultiplets.

In this paper, we consider unitary quiver gauge theories with generic real FI-parameters, whose Higgs branch $X$ is a smooth algebraic symplectic variety. We further assume that for generic real mass parameters, the fixed locus of corresponding $\mathbb{C}^*$ actions on $X$ consists of isolated fixed points. Under these assumptions, we will be able to determine the effective supersymmetric quantum mechanics exactly and compute the spaces of supersymmetric ground states, graded by a R-symmetries and global symmetries.

The Witten index of the effective supersymmetric quantum mechanics must reproduce limits of the supersymmetric twisted index on $S^1 \times \Sigma$. This can be computed by supersymmetric localisation, leading to elegant expressions involving JK residue formulae or a summations over solutions to Bethe equations [3–8]. This provides a non-trivial consistency check on our computations. Alternative localisation schemes that lead to geometric interpretation of the supersymmetric twisted index will underpin the constructions in this paper [9–11].

The spaces of supersymmetric ground states are closely related to the Hilbert space of the Rozansky-Witten topological twist [12] and its mirror on a Riemann surface $\Sigma$. In supersymmetric gauge theories, the relationship is subtle due to non-compactness of the target space. Our construction overcomes this difficulty by introducing real FI and mass parameters that break the R-symmetry necessary to perform a full $A$- and $B$-type topological twist on a generic three-manifold $M_3$ but are perfectly compatible on the specific background $M_3 = \mathbb{R} \times \Sigma$.

## 1.1 $A$-twist

In the $A$-twist, the $\mathcal{N} = 4$ supersymmetric quantum mechanics can roughly be understood as sigma model whose target space $M$ is the moduli space of twisted quasi-maps $\Sigma \to X$, where $X$ is the Higgs branch understood as an algebraic symplectic quotient.

In a supersymmetric gauge theory with compact connected gauge group $G$, the moduli space $M$ decomposes as a disjoint union of topologically distinct components $M_d$ labelled by the topological class $d \in \pi_1(G)$ of the $G$-bundle on $\Sigma$. Under some assumptions, each connected component $M_d$ is finite-dimensional and compact, but may be singular. In such cases, the supersymmetric ground states with topological charge $d \in \pi_1(G)$ are captured by a finite-dimensional $\mathcal{N} = 4$ supersymmetric quantum mechanics with target $M_d$.

The space of supersymmetric ground states is given by the hyper-cohomology

$$\mathcal{H}_A = \bigoplus_{d \in \pi_1(G)} \xi^d \, \mathbb{H}^\bullet(M_d, P_d), \tag{1}$$

where $P_d$ is a canonical perverse sheaf on $M_d$. The supersymmetric ground states are weighted by a formal parameter $\xi$ that keeps track of their charge under the topological or Coulomb branch global symmetry. The hyper-cohomology admits a pure Hodge structure or double grading that captures of the R-charges of supersymmetric ground states.

In many cases, especially when all the components of the degree $d \in \pi_1(G)$ is large compared to the genus $g$, the components $M_d$ are smooth. In this case, the hyper-cohomology reduces to the de Rham cohomology of $M_d$ with its Hodge decomposition, which recovers the supersymmetric ground states of a smooth $\mathcal{N} = 4$ supersymmetric sigma model.

For the purpose of computations, it is convenient to introduce real masses for flavour symmetries that act by isometries of the moduli spaces $M_d$. This corresponds to introducing a real

superpotential in the supersymmetric quantum mechanics, given by the moment map for the isometry of $M_d$. Under some assumptions, this is a perfect Morse-Bott function with critical loci of the form

$$\text{Sym}^{n_1}(\Sigma) \times \cdots \times \text{Sym}^{n_k}\Sigma \subset M_d \,, \tag{2}$$

where $(n_1, \ldots, n_k) \in \mathbb{Z}^k$ and $k$ is the rank of the gauge group. In such cases, the space of supersymmetric ground states may be computed explicitly from knowledge of the de Rham cohomology of symmetric products of the curve $\Sigma$.

## 1.2 $B$-twist

In the $B$-twist, the effective $\mathcal{N} = 4$ supersymmetric quantum mechanics involves vectormultiplets, hypermultiplet and twisted chiral multiplets. However, under some assumptions, the vectormultiplet and twisted hypermultiplet moduli spaces are lifted, leaving an $\mathcal{N} = 4$ supersymmetric quantum mechanics with hyper-Kähler target space given by the Higgs branch $X$.

The space of supersymmetric ground states is a priori given by $L^2$ harmonic forms on $X$ twisted by $g$ copies of exterior powers of the tangent bundle $T_X$. If $X$ were compact and the supersymmetric quantum mechanics gapped, this would have a cohomological description

$$\mathcal{H}_B = H^{0,\bullet}_{\bar{\partial}}(X, (\wedge^\bullet T_X)^g) \tag{3}$$

as in Rozansky-Witten theory [12].[1] However, the Higgs branch $X$ of a supersymmetric gauge theory is non-compact and correspondingly the supersymmetric quantum mechanics is not gapped, so this is subtle and a cohomological description is not immediately available.

In this paper, we introduce real mass parameters for Higgs branch global symmetry corresponding to tri-hamiltonian isometries of $X$. If the fixed locus of the isometry is compact, the spectrum of the supersymmetric quantum mechanics becomes gapped and a cohomological description opens up in terms of the cohomology of a Dolbeault operator, deformed by the moment map for the isometry. This has an algebraic description in terms of a Cousin spectral sequence, which captures instanton corrections to perturbative supersymmetric ground states localised around fixed loci [14–17].

Under some assumptions, the fixed locus consists of an isolated set of points $p \in X$. In this case, the supersymmetric ground states are built from Fock spaces attached to each fixed point together with potential instanton corrections. However, we argue that there are no instanton corrections with this amount of supersymmetry. The exact supersymmetric ground states in the $B$-twist are then given by

$$\mathcal{H}_B = \bigoplus_p \widehat{\text{Sym}}^\bullet V_p \,, \tag{4}$$

where $V_p$ is a vector space graded by $R$-symmetries and Higgs branch global symmetries encoding hypermultiplet fluctuations around the fixed point $p$. The hat denotes symmetric tensor powers, normalised by a factor of $(\det V_p)^{1/2}$.

## 1.3 Outline

The outline of the paper is as follows. In section 2, we summarise the general properties of spaces of supersymmetric ground states such as gradings by global and R-symmetries, which are determined by the supersymmetry algebra. In section 3 we summarise the class of supersymmetric gauge theories we consider and state clearly our assumptions. In sections 4 and 5 we construct the effective supersymmetric quantum mechanics and compute them in examples in the $A$-twist. In sections 6 and 7 we construct the effective supersymmetric quantum

---

[1]See also [13] for a related discussion.

mechanics and compute them in examples in the $B$-twist. In section 8, we discuss the matching of supersymmetric ground states under mirror symmetry. Finally, in section 9, we discuss relations to other work and potential future directions.

# 2 Cohomological Structures

In this section we outline homological structures involved in computing supersymmetric ground states of $\mathcal{N} = 4$ supersymmetric quantum mechanics that arise from 3d $\mathcal{N} = 4$ theories twisted on $\mathbb{R} \times \Sigma$. These constructions depend only on the supersymmetry algebra, the existence of global symmetries, and appropriate constraints to ensure a gapped spectrum. In sections 4 and 6, we will then construct supersymmetric ground states in the $A$-twist and $B$-twist that are compatible with these structures.

## 2.1 Twisted 1d $\mathcal{N} = 4$ Supersymmetry

We consider a 3d $\mathcal{N} = 4$ supersymmetric theory with R-symmetry $SU(2)_H \times SU(2)_C$ and global symmetry $G_H \times G_C$, where $G_H$ couples to vectormultiplet and $G_C$ to a twisted vectormultiplet. The supersymmetry algebra in euclidean $\mathbb{R}^3$ is

$$\{Q_\alpha^{A\dot{A}}, Q_\beta^{B\dot{B}}\} = \epsilon^{AB}\epsilon^{\dot{A}\dot{B}}P_{\alpha\beta} - \epsilon_{\alpha\beta}\epsilon^{AB}Z^{\dot{A}\dot{B}} - \epsilon_{\alpha\beta}\epsilon^{\dot{A}\dot{B}}Z^{AB}. \tag{5}$$

The central charges are

$$Z^{AB} = \zeta^{AB} \cdot J_C, \qquad Z^{\dot{A}\dot{B}} = m^{\dot{A}\dot{B}} \cdot J_H, \tag{6}$$

where $J_C$, $J_H$ denote Cartan generators of the global symmetry and $m^{\dot{A}\dot{B}}$ and $\zeta^{AB}$ are scalar expectation values for background vectormultiplets and twisted vectormultiplets respectively. In a supersymmetric gauge theory, they are mass and FI parameters respectively.

In what follows, we set $m^{\dot{1}\dot{1}} = 0$ and $\zeta^{11} = 0$ and rename the remaining real parameters by $m := m^{\dot{1}\dot{2}}$ and $\zeta := \zeta^{12}$ respectively. This fixes unbroken maximal tori $U(1)_H \times U(1)_C$ and $T_H \times T_C$ of the R-symmetry and global symmetry respectively.

We consider the twisted reduction of this supersymmetry algebra on $\mathbb{R} \times \Sigma$, where $\Sigma$ is a closed Riemann surface of genus $g$. The twist is performed using either the $U(1)_H$ or $U(1)_C$ R-symmetry to preserve an $\mathcal{N} = 4$ supersymmetric quantum mechanics on $\mathbb{R}$. We refer to these two choices as the $A$-twist and $B$-twist respectively. Starting from euclidean coordinates $\{x^1, x^2, x^3\}$ we twist in the $x^{1,2}$-plane, which is then replaced by $\Sigma$. This leads to a supersymmetric quantum mechanics in the $x^3$-direction.

### 2.1.1 A-Twist

There are four supercharges that commute with the diagonal combination of $U(1)_H$ and rotations in the $x^{1,2}$-plane, which are

$$Q^{\dot{A}} := Q_1^{1\dot{A}}, \qquad \widetilde{Q}^{\dot{A}} := Q_2^{2\dot{A}}. \tag{7}$$

After the twist, they are scalar on the $x^{1,2}$-plane and generate the 1d $\mathcal{N} = 4$ supersymmetry algebra

$$
\begin{aligned}
\{Q^{\dot{A}}, Q^{\dot{B}}\} &= 0, \\
\{Q^{\dot{A}}, \widetilde{Q}^{\dot{B}}\} &= \epsilon^{\dot{A}\dot{B}}(H - \zeta \cdot J_C) - m^{\dot{A}\dot{B}} \cdot J_H, \\
\{\widetilde{Q}^{\dot{A}}, \widetilde{Q}^{\dot{B}}\} &= 0,
\end{aligned}
\tag{8}
$$

where we have defined the Hamiltonian $H := P_3$. We note that the form of the Hamiltonian $H$ may also depend on the parameters $m$ and $\zeta$. The supercharges act on the Hilbert space of the supersymmetric quantum mechanics in such a way that $(Q^{\dot 1})^\dagger = \widetilde{Q}^{\dot 2}$ and $(Q^{\dot 2})^\dagger = -\widetilde{Q}^{\dot 1}$.

Recalling we set the complex mass parameter $m^{\dot 1 \dot 1} = 0$, it is convenient to decompose the supersymmetry algebra into a pair of commuting $\mathcal{N} = 2$ subalgebras with non-vanishing anti-commutators

$$
\begin{aligned}
\{Q_+, Q_+^\dagger\} &= H - \zeta \cdot J_C - m \cdot J_H, \\
\{Q_-, Q_-^\dagger\} &= H - \zeta \cdot J_C + m \cdot J_H,
\end{aligned}
\tag{9}
$$

where $Q_+ := Q^{\dot 1}$ and $Q_- := \widetilde{Q}^{\dot 1}$. It will also be useful to consider the diagonal $\mathcal{N} = 2$ subalgebra

$$
\{Q, Q^\dagger\} = 2(H - \zeta \cdot J_C)
\tag{10}
$$

generated by $Q := Q_+ + Q_-$. In the absence of the FI parameter $\zeta$, the combination $Q$ defines a fully topological $A$-twist or mirror Rozansky-Witten twist on $\mathbb{R} \times \Sigma$ that is compatible with the mass parameters $m$.

### 2.1.2 B-Twist

There are four supercharges commuting with the diagonal combination of $U(1)_C$ and rotations in the $x^{1,2}$-plane,

$$
Q^A := Q_1^{A\dot 1}, \qquad \widetilde{Q}^A := Q_2^{A\dot 2}.
\tag{11}
$$

After the twist, they are scalar on the $x^{1,2}$-plane and generate a 1d $\mathcal{N} = 4$ supersymmetry algebra

$$
\begin{aligned}
\{Q^A, Q^B\} &= 0, \\
\{Q^A, \widetilde{Q}^B\} &= \epsilon^{AB}(H - m \cdot J_H) - \zeta^{AB} \cdot J_C, \\
\{\widetilde{Q}^A, \widetilde{Q}^B\} &= 0,
\end{aligned}
\tag{12}
$$

where the supercharges again act on the Hilbert space of the supersymmetric quantum mechanics in such a way that $(Q^{\dot 1})^\dagger = \widetilde{Q}^{\dot 2}$ and $(Q^{\dot 2})^\dagger = -\widetilde{Q}^{\dot 1}$.

Recalling we set the complex FI parameter $\zeta^{11} = 0$, it is again convenient to decompose the supersymmetry algebra into a pair of commuting $\mathcal{N} = 2$ subalgebras with non-vanishing anti-commutators

$$
\begin{aligned}
\{Q_+, Q_+^\dagger\} &= H - m \cdot J_H - \zeta \cdot J_C, \\
\{Q_-, Q_-^\dagger\} &= H - m \cdot J_H + \zeta \cdot J_C,
\end{aligned}
\tag{13}
$$

where $Q_+ := Q^1$ and $Q_- := \widetilde{Q}^1$. It will also be useful to consider the diagonal $\mathcal{N} = 2$ subalgebra

$$
\{Q, Q^\dagger\} = 2(H - m \cdot J_H)
\tag{14}
$$

generated by $Q := Q_+ + Q_-$. In the absence of the mass parameter $m$, the combination $Q$ defines a fully topological $B$-twist or Rozansky-Witten twist on $\mathbb{R} \times \Sigma$ that is compatible with the FI parameters $\zeta$.

### 2.1.3 Comment on Notation

The above notation is designed so we can discuss both twists in parallel by interchanging $H \leftrightarrow C$ and $\zeta \leftrightarrow m$. In the remainder of this section, we will write formulae explicitly for the $A$-twist, with the understanding that those in the B-twist are obtained by performing the above substitution. It is useful to note that $Q = Q_1^{1\dot 1}$ is a common supercharge preserved by both twists.

## 2.2 Gradings

Let $\Omega$ denote the full Hilbert space of the effective $\mathcal{N} = 4$ supersymmetric quantum mechanics. This transforms as a unitary representation of the $R$-symmetry $U(1)_H \times U(1)_C$ and global symmetry $T_H \times T_C$ left unbroken by generic real mass and FI parameters. We discuss the R-symmetry first and the flavour symmetry second.

Let $R_H, R_C$ denote integer generators of $U(1)_H, U(1)_C$. We then have a $\mathbb{Z} \times \mathbb{Z}$ grading on $\Omega$ from the decomposition into eigenspaces of $R_H, R_C$. It is sometimes convenient to define the following combinations in the $A$-twist,

$$
\begin{aligned}
R_+ &:= \frac{1}{2}(R_C - R_H), \\
R_- &:= \frac{1}{2}(R_C + R_H),
\end{aligned}
\tag{15}
$$

which, by at most a constant half integer shift, also have integer eigenvalues. The notation is chosen such that $Q_\pm$ commutes with $R_\mp$, as summarised in table 1.

We now introduce yet another pair of combinations

$$
\begin{aligned}
F &:= R_C, \\
R &:= \frac{1}{2}(R_C - R_H),
\end{aligned}
\tag{16}
$$

defining an $\mathbb{Z} \times \frac{1}{2}\mathbb{Z}$ grading. For reasons discussed in more detail below, we refer to $F$ as the "primary" or "cohomological" grading and $R$ as the "secondary" grading. Correspondingly, we denote the contribution from a state in cohomological degree $f$ and secondary degree $r$ by

$$
t^r \mathbb{C}[-f],
\tag{17}
$$

where we use a formal parameter $t$ to keep track of the secondary grading. The weights of the supercharges are again summarised in table 1.

Table 1: Summary of supercharges weights under $R$-symmetry generators in the $A$-twist.

|       | $R_H$ | $R_C$ | $F$ | $R$ |
|-------|-------|-------|-----|-----|
| $Q_+$ | $-1$  | $1$   | $1$ | $0$ |
| $Q_-$ | $1$   | $1$   | $1$ | $1$ |
| $Q$   | $*$   | $1$   | $1$ | $*$ |

Let us now return to the global symmetry. This commutes with the R-symmetry so each weight space of the above double grading transforms as unitary representation of the unbroken global symmetry $T_H \times T_C$. The contribution from a state transforming with weight $(\gamma_H, \gamma_C) \in \mathrm{Hom}(T_H \times T_C, U(1))$ is denoted by

$$
x^{\gamma_H} \xi^{\gamma_C} \mathbb{C},
\tag{18}
$$

where we introduce formal parameters $(x, \xi) \in T_H \times T_C$.

Finally, it may happen that the cohomological grading $F = R_C$ is incompatible with interpreting $(-1)^F$ as the usual $\mathbb{Z}_2$ fermion number in three dimensions. In the $A$-twist, this happens because monopole operators are bosons but may have odd R-charge $f$. This is ameliorated by introducing a new R-symmetry of the form

$$
\widetilde{R}_C = R_C - \lambda \cdot J_C,
\tag{19}
$$

for some co-character $\lambda \in \mathrm{Hom}(U(1), T_C)$ and defining instead

$$
\begin{aligned}
F &:= \widetilde{R}_C \,, \\
R &:= \frac{1}{2}(\widetilde{R}_C - R_H) \,.
\end{aligned}
\tag{20}
$$

There is an analogous potential redefinition in the $B$-twist where a new R-symmetry $\widetilde{R}_H$ is formed by mixing with the global symmetry $T_H$. This kind of redefinition was discussed in the context of the Rozansky-Witten twist in [18].

## 2.3  Supersymmetric Ground States

### 2.3.1  Definitions

We are interested in the space $\mathcal{H}$ of supersymmetric ground states annihilated by all four super-charges $Q_+, Q_+^\dagger, Q_-, Q_-^\dagger$. This inherits gradings by $F$, $R$ and $T_H \times T_C$. From the supersymmetry relations (9) and unitarity, supersymmetric ground states satisfy

$$
E - \zeta \cdot \gamma_C = 0 \,, \qquad m \cdot \gamma_H = 0 \,,
\tag{21}
$$

where $E$ denotes the eigenvalue of $H$. Supersymmetric ground states in the $A$-twist are therefore uncharged under $T_H$ for generic mass parameters $m$. Similarly, supersymmetric ground states in the $B$-twist are uncharged under $T_C$ for generic FI parameters $\zeta$.

The space of supersymmetric ground states has a number of equivalent definitions that are useful in different circumstances. First, it is convenient to introduce an intermediate space of half-BPS states $\mathcal{H}_{1/2}$ annihilated by $Q_+$, $Q_+^\dagger$, which satisfy

$$
E - \zeta \cdot \gamma_C - m \cdot \gamma_H = 0 \,.
\tag{22}
$$

A consequence of this definition and the unitary bound arising from (10) is that states in $\mathcal{H}_{1/2}$ obey $m \cdot \gamma_H \geq 0$ and this inequality is saturated by supersymmetric ground states. Namely,

$$
\mathcal{H} = \mathcal{H}_{1/2} \cap \ker(m \cdot J_H) \,.
\tag{23}
$$

In other words, for generic mass parameters $m$, supersymmetric ground states are states in $\mathcal{H}_{1/2}$ that are uncharged under the global symmetry $T_H$.

### 2.3.2  Cohomological Construction

As usual in supersymmetric quantum mechanics, it is helpful to introduce a cohomological description of supersymmetric ground states. Let us assume that the spectrum of $H - \zeta \cdot J_C$ is gapped. This is typically a condition on the theory and the parameters $m$ and $\zeta$, which we discuss further in section 2.5.

By a standard argument, the space of supersymmetric ground states can then be identified with the cohomology of the supercharge $Q$ generating the diagonal subalgebra (10). In more detail, let $\Omega^f$ denote the space of states of cohomological degree $f$. Then

$$
\mathcal{H}^f = H^f(\Omega^\bullet, Q)
\tag{24}
$$

is the space of supersymmetric ground states of cohomological weight $f$. A downside of this construction is that $Q$ does not transform with a definite weight under $R$, so this does not immediately yield the secondary grading on supersymmetric ground states. To reproduce the secondary grading in a cohomological framework, there are various way to proceed.

One method is to note that $Q$ does preserve the filtration

$$F^r \Omega^f = \bigoplus_{r' \leq r} \Omega^{f,r'}, \tag{25}$$

where $\Omega^{f,r} \subset \Omega$ denotes states with cohomological weight $f$ and secondary weight $r$. It is straightforward to see from table 1 that the supercharge $Q$ is compatible with the filtration and defines a differential $Q: F^r \Omega^f \to F^r \Omega^{f+1}$. We can then pass to cohomology

$$F^r \mathcal{H}^f = H^f(F^r \Omega^\bullet, Q), \tag{26}$$

which is a filtration on the space of supersymmetric ground states. The secondary grading is then recovered from the associated graded of this filtration,

$$\mathcal{H}^{f,r} = \frac{F^r \mathcal{H}^f}{F^{r-1} \mathcal{H}^f}. \tag{27}$$

This can be rephrased in terms of the spectral sequence associated to this filtration. The first step is to note that the intermediate space $\mathcal{H}_{1/2}$ admits a cohomological description

$$\mathcal{H}_{1/2}^{f,r} = H^f(\Omega^{\bullet,r}, Q_+), \tag{28}$$

in which the secondary grading is manifest because $Q_+$ commutes with $R$. This is the $E_1$-page of the spectral sequence associated to the filtration (25) and abuts to the space of supersymmetric ground states. The secondary grading remains intact on each page of the spectral sequence and therefore recovers the secondary grading on $\mathcal{H}$.

Finally, we could simply restrict to states annihilated by $m \cdot J_H$ from the beginning. The subalgebras (9) and (10) act in the same way on this subspace and the space of supersymmetric ground states is the cohomology of any linear combination of $Q_\pm$ on states annihilated by $m \cdot J_H$. In particular, we can choose to represent supersymmetric ground states as

$$\mathcal{H}^{f,r} = H^f(\Omega^{\bullet,r} \cap \ker(m \cdot J_H), Q_+) \tag{29}$$

to manifest the secondary grading. Equivalently, upon on restriction to the kernel of $m \cdot J_H$, the aforementioned spectral sequence collapses at the $E_1$-page.

## 2.4 Recovering the Twisted Index

The supersymmetric twisted indices are defined as Witten indices of the $\mathcal{N} = 4$ supersymmetric quantum mechanics and may also be regarded as partition functions on $S^1 \times \Sigma$.

We start from the following expression in the $A$-twist,

$$\mathcal{I} = \text{Tr}_\Omega (-1)^F e^{-\beta H} e^{-i\beta a_R R} e^{-i\beta i a_C \cdot J_C} e^{-i\beta a_H \cdot J_H}, \tag{30}$$

where we have introduced constant background connections $a_R$, $a_H$, $a_C$ around $S^1$ for $R$, $T_H$, $T_C$ respectively and the circumference of the circle is $\beta$. A standard argument shows that this receives contributions only from the subspace $\mathcal{H}_{1/2}$ annihilated by $Q_+$, $Q_+^\dagger$. The index can therefore be expressed more succinctly as

$$\mathcal{I} = \text{Tr}_{\mathcal{H}_{1/2}} (-1)^F t^R x^{J_H} \xi^{J_C}, \tag{31}$$

where

$$\xi := e^{-\beta(\zeta + i a_C)}, \qquad x := e^{-\beta(m + i a_H)}, \qquad t := e^{-i\beta a_R}. \tag{32}$$

This notation is designed to be compatible with that introduced in section 2.2. Note that a redefinition of the R-charge as in equation (19) is implemented by $\xi \to (-t^{-\frac{1}{2}})^\lambda \xi$.

In the limit $t \to 1$, the twisted index only receive contributions from supersymmetric ground states in $\mathcal{H}$, which are not charged under the flavour symmetry $T_H$. We therefore find that the limit

$$\lim_{t \to 1} \mathcal{I} = \text{Tr}_{\mathcal{H}}(-1)^F \xi^{J_C} \tag{33}$$

counts supersymmetric ground states graded by $(-1)^F$ and $J_C$.

## 2.5 Parameter Dependence

The space of supersymmetric ground states $\mathcal{H}$ depends on various deformation parameters preserving the 1d $\mathcal{N} = 4$ supersymmetry algebra. This includes expectation values for background vectormultiplets and twisted vectormultiplets, such as mass parameters $m$, FI parameters $\zeta$, and certain background connections on $\Sigma$. This dependence is captured by a supersymmetric Berry connection. They type of supersymmetric Berry connection relevant in this context have been studied in references [19–22].

We content ourselves here with describing the dependence on the real parameters $m$, $\zeta$. The mass parameters $m$ are expectation values for the real scalar components of a background vectormultiplet for $T_H$. The supercharges depend on them in such a way that

$$\begin{aligned}
\partial_m Q_+ &= -[\mathcal{O}_H, Q_+], & \partial_m Q_+^\dagger &= +[\mathcal{O}_H, Q_+^\dagger], \\
\partial_m Q_- &= -[\mathcal{O}_H, Q_-], & \partial_m Q_-^\dagger &= +[\mathcal{O}_H, Q_-^\dagger],
\end{aligned} \tag{34}$$

where $\mathcal{O}_H$ is some operator in the quantum mechanics that plays the role of a moment map for the symmetry $T_H$. This means that $\partial_m + \mathcal{O}_H$ commutes with any linear combination of $Q_+$, $Q_-$ and induces a complex flat Berry connection on both $\mathcal{H}_{1/2}$ and the space of supersymmetric ground states $\mathcal{H}$. The conclusion is the same for the FI parameters $\zeta$.

This argument is only valid if the spectrum is gapped and $\mathcal{H}_{1/2}$ and $\mathcal{H}$ can be computed in cohomology. Under the assumptions to be outlined in section 3, the spectrum of the supersymmetric quantum mechanics is gapped provided the parameters $m$, $\zeta$ lie in the complement of certain hyperplanes. Namely,

$$\begin{aligned}
m &\in \mathfrak{t}_H - \bigcup_\lambda H_\lambda, \\
\zeta &\in \mathfrak{t}_C - \bigcup_\mu \widetilde{H}_\mu,
\end{aligned} \tag{35}$$

where

$$\begin{aligned}
H_\lambda &= \{m \in \mathfrak{t}_H \mid \langle \lambda, m \rangle = 0\}, \\
\widetilde{H}_\lambda &= \{t \in \mathfrak{t}_H \mid \langle \mu, t \rangle = 0\}
\end{aligned} \tag{36}$$

are hyperplanes labelled by weights $\lambda$, $\mu$ of $T_H$, $T_C$. A more precise statement is therefore that we obtain complex flat Berry connections on the complement of these hyperplanes.[2]

The hyperplanes typically cut the parameter spaces $\mathfrak{t}_H$, $\mathfrak{t}_C$ into chambers. Throughout this paper, we denote such a pair of chambers by $\mathfrak{c}_H$, $\mathfrak{c}_C$. In practise, the existence of a flat Berry connection means we can assign graded vector spaces $\mathcal{H}_{1/2}$ and $\mathcal{H}$ to each pair of chambers $\mathfrak{c}_H$, $\mathfrak{c}_C$ in the space of mass and FI parameters.

Let us now discuss how this parameter dependence translates to the twisted index. From the perspective of a path integral on $S^1 \times \Sigma$, the mass and FI parameters are complexified by

---

[2]If we were to introduce complex masses $m^{1i}$ and work with harmonic representatives of cohomology classes the Berry connection would lift to a solution of the generalised Bogomolnyi equations on $\mathfrak{t}_H \otimes \mathbb{R}^3$ with Dirac monopole singularities along codimension-three loci $H_\alpha \otimes \mathbb{R}^3$.

background connections $a_H$, $a_C$ around $S^1$. The twisted index $\mathcal{I}$ is then a rational function of the fugacities introduces in (32):

$$\xi = e^{-\beta(\zeta + ia_C)}, \qquad x = e^{-\beta(m + ia_H)}. \tag{37}$$

Let us compare this with the definition of the twisted index as a trace. We can imagine computing separate twisted indices for each pair of chambers,

$$\mathrm{Tr}_{\mathcal{H}^{1/2}_{\mathfrak{c}_H, \mathfrak{c}_C}} (-1)^F t^R x^{J_H} \xi^{J_C}, \tag{38}$$

which begin life as different formal Laurent series in $x$, $\xi$. However, as a consequence of the holomorphicity of the index, they are expansions of the same rational function $\mathcal{I}$ when the parameters are chosen such that $-\log|x| \in \mathfrak{c}_H$, $-\log|\xi| \in \mathfrak{c}_C$. Conversely, expanding the twisted index $\mathcal{I}$ in the region with parameters $-\log|x| \in \mathfrak{c}_H$, $-\log|\xi| \in \mathfrak{c}_C$ will reproduce the trace over $\mathcal{H}^{1/2}$ in the chamber $\mathfrak{c}_H$, $\mathfrak{c}_C$.

Finally, we have seen that supersymmetric ground states $\mathcal{H}$ are those states in $\mathcal{H}_{1/2}$ that are annihilated by $m \cdot J_H$. Combining this with the above paragraph provides a way to gain information on the secondary grading of supersymmetric ground states from the twisted index. In particular, let us consider the limit $m \to \infty$ in the $A$-twist with $m \in \mathfrak{c}_H$. This corresponds to

$$x^\lambda \to \begin{cases} 0 & \langle \lambda, m \rangle > 0 \quad \text{for all} \quad m \in \mathfrak{c}_H \\ \infty & \langle \lambda, m \rangle < 0 \quad \text{for all} \quad m \in \mathfrak{c}_H \end{cases}. \tag{39}$$

In this limit

$$\lim_{\mathfrak{c}_H} \mathcal{I} = \mathrm{Tr}_{\mathcal{H}_{\mathfrak{c}_H}} (-1)^F t^R \xi^{J_C}, \tag{40}$$

which receives contributions only from supersymmetric ground states in the chamber $\mathfrak{c}_H$. This will provide a useful consistency check on the secondary grading of supersymmetric ground states.

# 3 Supersymmetric vacua and Assumptions

A 3d $\mathcal{N} = 4$ supersymmetric gauge theory is specified by compact gauge group $G$ together with a linear quaternionic representation, which we assume of the form $N := T^*M$ with $G \subset USp(M)$. An example are unitary quiver gauge theories, where $G = \prod_I U(n_I)$ and $T^*M$ is built from fundamental and bifundamental representations of the factors.

The theory has an abelian topological global symmetry $T_C = \mathrm{Hom}(\pi_1(G), U(1))$, which may be enhanced in the IR to a non-abelian symmetry $G_C$. In addition, there is a global symmetry $G_H$ acting on the hypermultiplets,

$$G_H = N_{USp(M)}(G)/G. \tag{41}$$

The mass and FI parameters correspond to constant expectation values for background vectormultiplets and twisted vectormultiplets respectively. As in section 2, we restrict here to real parameters

$$m := m^{\dot{1}\dot{2}}, \quad \zeta := \zeta^{12}, \tag{42}$$

with the remaining parameters to zero when not otherwise specified. We assume these parameters are generic and break the flavour and R-symmetries to their respective maximal tori $T_C$, $T_H$, $U(1)_H$, $U(1)_C$.

Correspondingly, we decompose the hypermultiplet scalars $X^A$ into complex components $(X, Y)$ and the vectormultiplet scalars $\sigma^{\dot{A}\dot{B}}$ into real and complex components $\sigma$, $\varphi$, $\varphi^\dagger$. The

Table 2: Fields and charges

|       | $G$   | $R_H$ | $R_C$ |
|-------|-------|-------|-------|
| $\sigma$ | Adj   | 0     | 0     |
| $\phi$   | Adj   | 0     | +2    |
| $X$      | $M$   | +1    | 0     |
| $Y$      | $M^*$ | −1    | 0     |

charges of these fields under the unbroken maximal torus of the R-symmetry are shown in table 2.

Such theories are endowed with an intricate moduli space of vacua that may include Higgs, Coulomb and mixed branches. Of particular importance to this paper is the Higgs branch, which we denote by $X$. This receives no quantum corrections and can be determined classically. It takes the form of a hyper-Kähler or algebraic symplectic quotient.

Let us first set $m = 0$. Then the classical vacuum equations are

$$
\begin{aligned}
\mu_{\mathbb{R}} = \zeta, && [\varphi, \varphi^{\dagger}] = 0, \\
\mu_{\mathbb{C}} = 0, && [\sigma, \varphi] = 0, \\
\sigma \cdot X = 0, && \sigma \cdot Y = 0, \\
\varphi \cdot X = 0, && \varphi \cdot Y = 0,
\end{aligned}
\tag{43}
$$

where vectormultiplet scalars act in the appropriate representation and

$$
\mu_{\mathbb{R}} = X \cdot X^{\dagger} - Y^{\dagger} \cdot Y, \quad \mu_{\mathbb{C}} = X \cdot Y
\tag{44}
$$

are the real and complex moment maps for the $G$ action on $T^* M$. In writing the real moment map equation, we identify $\zeta$ with an element of $\mathfrak{g}^*$ through $\mathfrak{t}_C \cong Z(\mathfrak{g}^*) \subset \mathfrak{g}^*$.

## 3.1 Assumptions

In this paper, we will assume that for generic values of the FI parameter $\zeta$, the gauge symmetry is broken to at most a discrete subgroup and therefore $\sigma = \varphi = 0$. This means $\mu_{\mathbb{C}}^{-1}(0) \cap \mu_{\mathbb{R}}^{-1}(\zeta) \subset T^* M$ has no continuous stabilisers for generic $\zeta$. This is a constraint on the data $G$, $M$.

If we restrict attention to unitary quivers, discrete stabilisers cannot appear (see e.g. [23], section 4). The assumption is then equivalent to the statement that the gauge symmetry is completely broken or $\mu_{\mathbb{C}}^{-1}(0) \cap \mu_{\mathbb{R}}^{-1}(\zeta)$ has no non-trivial stabilisers. Although not strictly necessary for this paper, to avoid some technicalities we restrict attention to this case.

With this understood, the remaining equations in (43) describe the Higgs branch as a smooth hyper-Kähler quotient,

$$
X \cong \mu_{\mathbb{C}}^{-1}(0) \cap \mu_{\mathbb{R}}^{-1}(\zeta)/G,
\tag{45}
$$

which is a Nakajima quiver variety. In our assumption, a generic FI parameter means

$$
\zeta \in \mathfrak{t}_C - \bigcup_{\mu} \widetilde{H}_{\mu},
\tag{46}
$$

where the real hyperplanes $\widetilde{H}_{\mu} \subset \mathfrak{t}_C$ correspond to values of the real FI parameter where there is an unbroken gauge symmetry and non-trivial stabilisers. The hyperplanes split the parameter space into chambers as in section 2.5. We typically fix a chamber $\zeta \in \mathfrak{c}_C$.

In this paper, it is convenient to introduce an alternative description of the Higgs branch as an algebraic symplectic quotient

$$X \cong \mu_{\mathbb{C}}^{-1}(0) /\!/_\zeta G_{\mathbb{C}}, \tag{47}$$

where the real moment map equation is replaced by a stability condition depending in a piecewise constant manner on the FI parameter $\zeta$ and the quotient is now by complex gauge transformations. The stability condition and the resulting smooth algebraic symplectic variety $X$ depends only on the chamber $\mathfrak{c}_H$. Our assumption can then be summarised as follows:

- **Assumption I**: for generic a FI parameter $\zeta \in \mathfrak{c}_H$, the Higgs branch $X$ is a smooth algebraic symplectic variety.

Let us now introduce a real mass parameters $m$, which replaces $\sigma \to \sigma + m$ in the vacuum equations (43). From an algebraic point of view, the mass parameters generate a $\mathbb{C}_m^* \subset T_{H,\mathbb{C}}$ action on $X$ preserving the holomorphic symplectic form and the solutions of the vacuum equations now correspond to fixed loci of this action. We will further assume that for generic mass parameters $m$, the fixed locus is a set of isolated points, which will abstractly index by $I$

$$X^{\mathbb{C}_m^*} = \bigsqcup_I \{p_I\}. \tag{48}$$

This is again a condition on the data $G, M$. In our assumption, a generic mass parameter means

$$m \in \mathfrak{t}_H - \bigcup_\lambda H_\lambda, \tag{49}$$

where the real hyperplanes $H_\lambda \subset \mathfrak{t}_H$ correspond to mass parameters where the $\mathbb{C}^*$ action no longer has isolated fixed points. They can be described explicitly as

$$H_\lambda = \{m \in \mathfrak{t}_H \mid \langle \lambda, m \rangle = 0\}, \tag{50}$$

where $\lambda$ runs over all weights in the $T_H$ weight decompositions of the tangent space $T_p X$ for all fixed points $p$. The hyperplanes split the parameter space into chambers as in section 2.5. We typically fix a chamber $\zeta \in \mathfrak{c}_C$. Our assumption can be summarised as follows:

- **Asssumption II**: for generic mass parameters $m \in \mathfrak{c}_H$, the corresponding $\mathbb{C}_m^*$ action on the Higgs branch $X$ has isolated fixed points.

Let us illustrate these assumptions in the case of supersymmetric QCD with $G = U(k)$ and $N$ hypermultiplets in the fundamental representation. In this case the flavour symmetries are $T_H = U(1)^N/U(1)$ and $T_C = U(1)$ and we can turn on real mass parameters $(m_1, \ldots, m_N)$ with $\sum_j m_j = 0$ and a real FI parameter $\zeta$.

This satisfies assumptions I and II provided $N \geq k$. First, the Higgs branch is then smooth and isomorphic to $X \cong T^* G(k, N)$ whenever $\zeta \neq 0$. It therefore therefore satisfies Assumption I with two chambers $\mathfrak{c}_C = \{\zeta > 0\}$ and $\mathfrak{c}_C = \{\zeta < 0\}$. Second the $\mathbb{C}_m^*$ action generated by generic mass parameters with $m_i \neq m_j$ for $i \neq j$ also has isolated fixed points. The theory therefore satisfies Assumption II with $N!$ chambers $\mathfrak{c}_H$ corresponding to orderings of $N$ distinct mass parameters. This is a little weaker than the theory being good, which requires $N \geq 2k$ [24].

Other examples of unitary quiver gauge theories satisfying both assumptions I and II are $T[SU(N)]$ and the ADHM quiver.

## 3.2 Fixed Points

Under assumptions I and II, we can give a more concrete description of the isolated fixed points. This description is related to the Jeffrey-Kirwan prescription for the supersymmetric twisted index, which is familiar in the context of supersymmetric localisation computations.

Notice that at a fixed point, the gauge group must be completely broken. This means in particular that the vacuum equations (43) in the presence of a real mass $m$ (that is with the substitution $\sigma \mapsto \sigma + m$) must uniquely fix $\sigma$. This requirement is equivalent to the choice of a set of $k := \mathrm{rank}(G)$ weights $\{\rho_1, \ldots \rho_k\} \in \mathfrak{t}^*$ such that

- only hypermultiplet scalars transforming with these weights are non-vanishing;

- the set of weights $\{\rho_1, \ldots \rho_k\}$ must span $\mathfrak{t}^*$.

Furthermore, the real moment-map equation implies that

- the positive cone of these set of weights must contain the FI parameter $\zeta$.

$$\zeta \in \mathrm{Cone}^+(\{\rho_1, \ldots \rho_k\}). \tag{51}$$

This corresponds to the data of a non-degenerate, projective singularity that enters the definition of the Jeffrey-Kirwan residue prescription[3]. Finally, since we require the absence of discrete stabilisers, the square-matrix formed by the set of weights must be unimodular. These properties will be important for the computation of supersymmetric ground states, see in particular sections 5.4 and 7.4.

## 3.3 Tangent weights

Let us finally discuss the weight decomposition of the tangent space $T_p X$ of the Higgs branch at a fixed point $p$. We keep track of the weights in the manner introduced in section 2.2. First, for a given mass parameter $m$ there is a decomposition

$$T_p X = N_p^+ \oplus N_p^- \tag{52}$$

into positive and negative weight spaces for the corresponding $\mathbb{C}_m^*$ action. This decomposition depends only on the chamber $\mathfrak{c}_H$. The weights $\lambda$ or summands $x^\lambda \mathbb{C}$ appearing in the $T_H$ weight decomposition of $N_\pm$ obey $\pm\langle \lambda, m \rangle > 0$. Second, the algebraic symplectic form $\Omega$ on $X$ transforms with degree

$$F(\Omega) = \begin{cases} 0 & A\text{-twist} \\ 2 & B\text{-twist} \end{cases}, \qquad R(\Omega) = \begin{cases} -1 & A\text{-twist} \\ +1 & B\text{-twist} \end{cases}, \tag{53}$$

which implies that

$$(N_p^+)^\vee = \begin{cases} t^{-1} N_p^- & A\text{-twist} \\ t N_p^-[-2] & B\text{-twist} \end{cases}. \tag{54}$$

This will play an important role in our construction of the space of supersymmetric ground states in subsequent sections.

---

[3]In addition, notice that since automorphisms of the gauge group are ruled out at the fixed points, in our assumptions the various components of $\sigma$ cannot coincide. In terms of supersymmetric localisation computations of the twisted index, these would correspond to vectormultiplet poles.

# 4 Localisation in the *A*-Twist

The aim of this section is use supersymmetric localisation to reduce the *A*-twisted theory on $\mathbb{R} \times \Sigma$ to an explicit $\mathcal{N} = 4$ supersymmetric quantum mechanics that captures the space of supersymmetric ground states $\mathcal{H}$. The method is supersymmetric localisation. Following our previous work [9–11], we will choose a Higgs branch type localisation scheme leading to an algebro-geometric interpretation of the space of supersymmetric ground states.

## 4.1 Decomposing Supermultiplets

In the *A*-twist, 3d $\mathcal{N} = 4$ supermultiplets decompose into 1d $\mathcal{N} = (2, 2)$ supermultiplets. A 3d $\mathcal{N} = 4$ gauge theory of the type introduced in section 3 can be regarded as an infinite-dimensional gauged supersymmetric quantum mechanics as follows.

First, let $P$ denote a principal *G*-bundle on $\Sigma$ with connection *A*. Then we have the following multiplets in the supersymmetric quantum mechanics:

- A 1d $\mathcal{N} = (2, 2)$ vectormultiplet for the infinite-dimensional group of gauge transformations or automorphisms of $P$. The bosonic components are $A_3$, $\sigma$, $\varphi$, and an auxiliary field

$$D_{1d} := D - i * F_A,\tag{55}$$

  where $F_A$ is the curvature of *A* and $*$ is the Hodge star operator on $\Sigma$.

- A 1d $\mathcal{N} = (2, 2)$ chiral multiplet $\bar{\partial}_A$ parametrising the complex structure on vector bundles associated to $P$. In local coordinates $(z, \bar{z})$ on $\Sigma$, the bosonic component is $A_{\bar{z}}$.

- 1d $\mathcal{N} = (2, 2)$ chiral multiplets $(X, Y)$ transforming as sections of $S \otimes (P \times_G T^*M)$, where $S$ is a spin structure on $\Sigma$.

A crucial ingredient is a 1d $\mathcal{N} = (2, 2)$ superpotential

$$W = \int_{\Sigma} X \bar{\partial}_A Y,\tag{56}$$

which incorporates kinetic terms for the chiral multiplets along $\Sigma$ and the complex moment map constraint from the perspective of the supersymmetric quantum mechanics.

## 4.2 Localisation to Vortices

To perform supersymmetric localisation, it is convenient to use 1d $\mathcal{N} = (2, 2)$ supersymmetric Lagrangians corresponding to the three bullet points above. Let $L_V$, $L_C$ and $L_W$ denote exact Lagrangians for the 1d $\mathcal{N} = 4$ vectormultiplet, chiral multiplet and superpotential.

We will also need to decompose the 3d FI parameter $\zeta$ into language of $\mathcal{N} = (2, 2)$ supersymmetric quantum mechanics. The Lagrangian is given by

$$\begin{aligned} L_{FI} &= \frac{i\zeta}{2\pi} D \\ &= \frac{i\zeta}{2\pi} D_{1d} + \frac{\zeta}{2\pi} * F_A, \end{aligned}\tag{57}$$

where we have used the relation (55) between the vectormultiplet auxiliary fields. The first summand in the second line is a 1d FI parameter. The second is a coupling between $\zeta$ to the supersymmetric generator of the topological symmetry and is responsible for the grading by the topological symmetry. We will denote these two terms by $L_{FI,1d}$ and $L_{\zeta}$ respectively. The

1d FI parameter $L_{FI,1d}$ is exact with respect to the combinations $Q_+ + Q_+^\dagger$ and $Q_- + Q_-^\dagger$. On the other hand, the coupling $L_\zeta$ is not exact.

Finally, we will need to introduce a Lagrangian $L_m$ for mass parameters, by coupling to a background 1d $\mathcal{N} = (2, 2)$ vectormultiplet and turning on expectation values for the real scalar. As with $L_\zeta$, the Lagrangian $L_m$ is not exact.

Our starting point for supersymmetric localisation is

$$L = \frac{1}{t^2}\left(\frac{1}{e^2}L_V + L_C + L_{FI,1d}\right) + \frac{1}{g^2}L_W + L_\zeta + L_m \,, \tag{58}$$

where we have introduced positive constants $t^2$, $g^2$ in front of linear combinations of exact Lagrangians. The notation $e^{-2}$ is shorthand for a fixed inner product $\mathfrak{g} \times \mathfrak{g} \to \mathbb{R}$ appearing in the vectormultiplet Lagrangian. Provided the supersymmetric quantum mechanics remains gapped, we can scale the parameters $t^2$, $g^2$ to compute the space of supersymmetric ground states.

Let us first set the mass parameters to vanish, $m = 0$. In the limit $t^2$, $g^2 \to 0$, the action is minimised by solutions of the following system of equations on $\Sigma$,

$$\frac{1}{e^2} * F_A + \mu_{\mathbb{R}} = \zeta \,, \qquad [\sigma^{\dot{A}\dot{B}}, \sigma^{\dot{C}\dot{D}}] = 0 \,, \qquad d_A \sigma^{\dot{A}\dot{B}} = 0 \,,$$

$$\sigma^{\dot{A}\dot{B}} \cdot X = 0 \,, \qquad \sigma^{AB} \cdot Y = 0 \,,$$

$$\bar{\partial}_A X = 0 \,, \qquad \bar{\partial}_A Y = 0 \,, \qquad \mu_{\mathbb{C}} = 0 \,. \tag{59}$$

In the absence of real mass parameters, it is convenient to use $SU(2)_C$ covariant notation $\sigma^{\dot{A}\dot{B}}$ for the vectormultiplet scalars. The first two lines arise from saddle points of the combination multiplying $t^{-2}$ and the final line from the superpotential contribution multiplying $g^{-2}$.

Under Assumption 1 in section 3 and assuming $\zeta$ is chosen generically in some chamber, solutions will completely break the gauge symmetry and $\sigma^{\dot{A}\dot{B}}$ vanishes identically. We may then focus on solutions of the symplectic vortex equations

$$\frac{1}{e^2} * F_A + \mu_{\mathbb{R}} = \zeta \,,$$

$$\bar{\partial}_A X = 0 \,, \qquad \bar{\partial}_A Y = 0 \,, \qquad \mu_{\mathbb{C}} = 0 \,. \tag{60}$$

It is now convenient to introduce a dimensionless parameter

$$\widetilde{\zeta} := \frac{e^2 \text{vol}(\Sigma)}{2\pi} \zeta \in \mathfrak{t} \,, \tag{61}$$

where we view the gauge coupling here as a map $e^2 : \mathfrak{g}^* \to \mathfrak{g}$. Integrating the first equation in (60) over $\Sigma$ leads to a number of important conclusions. First, if $\widetilde{\zeta}$ is a co-character of $G$ then there will be additional Coulomb branch solutions with unbroken gauge symmetry. Second, at finite $\widetilde{\zeta}$, there is a bound on the degree or vortex number. These conclusions are related to the wall-crossing phenomena studied in [10].

In this paper, we therefore want to avoid these phenomena by passing to the infinite-tension limit. Concretely, this is the limit

$$|\widetilde{\zeta}| \to \infty \,, \tag{62}$$

with $\zeta \in \mathfrak{c}_C$ in a fixed chamber. This can be regarded as an infrared or strong coupling limit implemented by $e^2 \text{vol}(\Sigma) \to \infty$ with fixed $\zeta \in \mathfrak{c}_H$. This limit is important to obtain supersymmetric ground states that can be mapped under mirror symmetry.

To determine the effective supersymmetric quantum mechanics, one must understand the moduli space of solutions (60) and determine the massless fluctuations of all fields around solutions. In the following, we consider a formal approach to the moduli space as an infinite-dimensional quotient, before introducing a concrete finite-dimensional model to perform computations.

## 4.3 An Infinite-dimensional Model

Let us now consider massless fluctuations around a solution of the symplectic vortex equations (60). For the bosonic fields, this is done by linearising the equations around a solution. For fermions, one can expand Yukawa couplings around a solution to determine the massless fermions.

To simplify our notation, let us define

$$
\begin{aligned}
P_X &:= S \otimes P_M \,, \\
P_Y &:= S \otimes P_{M^*} \,,
\end{aligned}
\tag{63}
$$

where $P_R := P_{\mathfrak{g}} \times_G R$ is an associated vector bundle in representation $R$ and $S$ is a choice of spin structure, $S^2 \cong K_\Sigma$. The vector bundles $P_R$ are equipped with a hermitian metric from the inner product $e^{-2}$ in the vectormultiplet Lagrangian. This extends to a hermitian metric on $P_X$, $P_Y$ by combining with a hermitian metric on $S$.

With this notation in hand, it was shown in our previous work [9] that the massless fluctuations around a solution of the generalised vortex equations (60) are encoded in the cohomology of the following complex

$$
\Omega^0(P_{\mathfrak{g}}) \xrightarrow{\alpha^0}
\begin{array}{c}
t^{\frac{1}{2}}\Omega^0(P_X) \oplus t^{\frac{1}{2}}\Omega^0(P_Y) \\
\oplus \\
\Omega^1(P_{\mathfrak{g}})
\end{array}
\xrightarrow{\alpha^1}
\begin{array}{c}
t\Omega^0\left(K_\Sigma \otimes P_{\mathfrak{g}^*}\right) \\
\oplus \\
t^{\frac{1}{2}}\Omega^1(P_Y) \oplus t^{\frac{1}{2}}\Omega^1(P_X)
\end{array}
\xrightarrow{\alpha^2} t\Omega^1\left(K_\Sigma \otimes P_{\mathfrak{g}^*}\right) , \tag{64}
$$

where each summand represents a supermultiplet for the 1d $\mathcal{N}=2$ subalgebra generated by $Q_+$. The summands in the complex are given explicitly as follows:

- The $\mathcal{N}=(2,2)$ vectormultiplet has been decomposed into a field strength Fermi multiplet $\Omega^0(P_{\mathfrak{g}})$ generating infinitesimal gauge transformations and a chiral multiplet $\Omega^1(K_\Sigma \otimes P_{\mathfrak{g}^*})$ parametrising fluctuations of $\varphi$ [4].

- The $\mathcal{N}=(2,2)$ chiral multiplet $\bar{\partial}_A$ has been decomposed into a chiral multiplet $\Omega^1(P_{\mathfrak{g}})$ and a Fermi multiplet $\Omega^0(K_\Sigma \otimes P_{\mathfrak{g}^*})$.

- The $\mathcal{N}=(2,2)$ chiral multiplet $X$ has been decomposed into a chiral multiplet $\Omega^0(P_X)$ and a Fermi multiplet $\Omega^1(P_X)$. Similarly for $Y$.

The cohomological grading is represented horizontally here with the complex concentrated in degrees $F = -1, 0, 1, 2$, while the secondary grading represented as in section (2.2) by powers $t^R$. In particular, 1d $\mathcal{N}=2$ chiral multiplets appear in even cohomological degrees $F = 0, 2$ and Fermi multiplets in odd degree $F = -1, 1$.

The differentials $\alpha^0$, $\alpha^1$, $\alpha^2$ in the complex are given by infinitesimal gauge transformations and derivatives of the superpotential $W$. Explicitly,

$$
\alpha^0 : \lambda \longrightarrow
\begin{array}{c}
\lambda \cdot X \oplus \lambda \cdot Y \\
\oplus \\
\bar{\partial}_A \lambda
\end{array} , \tag{65}
$$

$$
\alpha^1 \quad
\begin{array}{c}
\delta X \oplus \delta Y \\
\oplus \\
\delta \bar{A}
\end{array}
\longrightarrow
\begin{array}{c}
\delta X \cdot Y + X \cdot \delta Y \\
\oplus \\
(\bar{\partial}_A \delta Y + \delta \bar{A} \cdot Y) \oplus (\bar{\partial}_A \delta X + \delta \bar{A} \cdot X)
\end{array} , \tag{66}
$$

---

[4]This more accurately corresponds to fluctuations of the Hodge dual $* \varphi^\dagger$.

$$\alpha^2 : \quad \begin{matrix} \Lambda \\ \oplus \\ \eta_X \oplus \eta_Y \end{matrix} \quad \longrightarrow X \cdot \eta_X + Y \cdot \eta_Y + \bar{\partial}_A \Lambda \ . \tag{67}$$

It is straightforward to check that this is indeed a complex, namely $\alpha^0 \circ \alpha^1 = \alpha^1 \circ \alpha^2 = 0$, on solutions of the generalised vortex equations (60).

This is a standard deformation-obstruction complex for the symplectic vortex equations (60). It can be regarded as representing the tangent complex $T_{\mathfrak{M}}$ of a derived moduli space $\mathfrak{M}$ parametrising solutions to the generalised vortex equations modulo gauge transformations.

The complex has an important symmetry as a consequence of the isomorphism

$$\Omega^q(E) \cong \Omega^{1-q}(K_\Sigma \otimes E^*), \tag{68}$$

using the Hodge star operator and hermitian metric on a vector bundle $E$. Namely, if one takes the dual or cotangent complex and applies this isomorphism, one recovers the original tangent complex but shifted in cohomological and secondary degree,

$$T_{\mathfrak{M}}^\vee \cong t^{-1} T_{\mathfrak{M}}[1]. \tag{69}$$

In particular, this means that $\mathfrak{M}$ is $(-1)$-shifted symplectic with respect to the cohomological grading $F$. This structure is a general feature of 1d $\mathcal{N} = (2,2)$ quantum mechanics and will play an important role in constructing the space of supersymmetric ground states.

A useful way to understand this fact is to realise $\mathfrak{M}$ as a derived critical locus. This origin of this picture is the description of vortex moduli spaces as infinite-dimensional quotients [25] and more specifically in the context of 3d $\mathcal{N} = 4$ supersymmetric gauge theories [26, 27]. The starting point is the infinite-dimensional affine space

$$\mathcal{F} = \Omega^0(P_X) \oplus \Omega^0(P_Y) \oplus \Omega^1(P_{\mathfrak{g}}), \tag{70}$$

parametrised by the top components of the chiral multiplets $X$, $Y$, $\bar{\partial}_A$. This is equipped with a flat Kähler metric using the standard inner product on forms and the hermitian metric on the bundles $P_X$, $P_Y$, $P_{\mathfrak{g}}$. This space is acted on by the infinite-dimensional group of gauge transformations $\mathrm{Aut}(P)$ with moment map

$$\frac{1}{e^2} * F_A + \mu_{\mathbb{R}}(X, Y). \tag{71}$$

The superpotential $W$ is invariant under gauge transformations and descends to a function on the infinite-dimensional Kähler quotient. The moduli space $\mathfrak{M}$ is then identified with the derived critical locus of the superpotential $W$ on the Kähler quotient.

This construction is of course infinite-dimensional and so perhaps unsuitable for a rigorous definition of the space of supersymmetric ground states as it stands. We now introduce a finite-dimensional algebro-geometric model of $\mathfrak{M}$.

## 4.4 A Finite-Dimensional Model

We first note that there is a decomposition

$$\mathfrak{M} = \bigsqcup_{d \in \pi_1(G)} \mathfrak{M}_d \tag{72}$$

as a disjoint union of components labelled by the topological degree $d \in \pi_1(G)$ of the gauge bundle. Under Assumption I and for generic values of $\zeta$, this is expected to be a derived scheme.

We now introduce a finite-dimensional algebro-geometric description of each component $\mathfrak{M}_d$ using a Hitchin-Kobayashi type correspondence. Namely, $\mathfrak{M}_d$ has an algebraic description parametrising

1. a holomorphic $G_{\mathbb{C}}$-bundle $E$ of degree $d$,

2. a holomorphic section $(X, Y)$ of $K_{\Sigma}^{1/2} \otimes (E \times_G T^*M)$ subject to $\mu_{\mathbb{C}}(X, Y) = 0$,

subject to a stability condition depending in a piecewise constant way on $\widetilde{\zeta}$. As mentioned above, we consider the infinite-tension limit $|\widetilde{\zeta}| \to \infty$, with fixed $\zeta \in \mathfrak{c}_C$. Then $\mathfrak{M}_d$ is the derived moduli space parametrising $\widetilde{\zeta}$-stable twisted[5] quasi-maps $\Sigma \to X$ of degree $d$. The study of quasi-maps to GIT quotients has been pioneered in and their application to enumerative geometry explored in. The particular instance of twisted quasi-maps to algebraic symplectic quotients described above was introduced in [28].

From an algebraic perspective, massless fluctuations around a solution $(E, X, Y)$ are given by the cohomology of the complex

$$H^0(E_{\mathfrak{g}}) \longrightarrow \begin{array}{c} t^{\frac{1}{2}}H^0(E_X) \oplus t^{\frac{1}{2}}H^0(E_Y) \\ \oplus \\ H^1(E_{\mathfrak{g}}) \end{array} \longrightarrow \begin{array}{c} tH^0\left(K_{\Sigma} \otimes E_{\mathfrak{g}^*}\right) \\ \oplus \\ t^{\frac{1}{2}}H^1(E_X) \oplus t^{\frac{1}{2}}H^1(E_Y) \end{array} \longrightarrow tH^1\left(K_{\Sigma} \otimes E_{\mathfrak{g}^*}\right) \ . \quad (73)$$

The interpretation of the various summands in terms of chiral and Fermi multiplets is identical to that in equation (64). The difference is we now parametrise holomorphic sections from the outset, so the summands are finite-dimensional vector spaces and the differentials no longer involve covariant derivative $\bar{\partial}_A$. This construction can be globalised using a universal construction as in [28] to give the tangent complex of $\mathfrak{M}_d$.

A downside of this construction is that $\mathfrak{M}_d$ can no longer be realised as a derived critical locus on global quotient in a finite-dimensional setting. Nevertheless, as a consequence of Serre duality

$$H^0(E) \cong H^1(K_{\Sigma} \otimes E^*)^* \quad (74)$$

it remains true that

$$T_{\mathfrak{M}}^{\vee} \cong t^{-1} T_{\mathfrak{M}}[1] \quad (75)$$

and the derived moduli space is $(-1)$-shifted symplectic. This implies that the bosonic or classical truncation $M_d$ (obtaining by discarding the Fermi multiplet fluctuations) has a symmetric obstruction theory. This is the symmetric obstruction theory introduced in [28].

This leads to a natural proposal for the space of supersymmetric ground states, following a common theme in the realm of categorification of enumerative invariants. The $(-1)$-shifted symplectic structure ensures the existence of a canonical perverse sheaf $P_d$ on the classical truncation $M_d$. The proposal is then that the space of supersymmetric ground states coincides with the hyper-cohomology

$$\mathcal{H} = \sum_{d \in \pi_1(G)} \xi^d \, \mathbb{H}^{\bullet}(M_d, P_d). \quad (76)$$

The topological symmetry grading is manifest in this formula. The cohomological and secondary gradings arise from the Hodge structure on this hyper-cohomology.

To gain some familiarity, let us explain how it reproduces the expected result from supersymmetric quantum mechanics when the moduli space is smooth. Suppose that

$$\mathfrak{M}_d = T^*[-1]M_d, \quad (77)$$

where $M_d$ is a smooth projective variety. This corresponds to an $\mathcal{N} = (2, 2)$ supersymmetric quantum mechanics which is a smooth sigma model with target $M_d$. In this case, the perverse sheaf in question is the constant sheaf with shifted cohomological degree,

$$P_d = \underline{\mathbb{C}}_{M_d}[\dim M_d]. \quad (78)$$

---

[5]The adjective twisted refers to the $K_{\Sigma}^{1/2}$ appearing in the definition of the holomorphic sections.

Using the standard resolution of the constant sheaf by the de Rham complex

$$0 \to \underline{\mathbb{C}}_X \to \Omega_X^0 \to \Omega_X^1 \to \cdots, \tag{79}$$

we find

$$
\begin{aligned}
\mathcal{H} &= \sum_{d \in \pi_1(G)} \xi^d \, \mathbb{H}^\bullet(M_d, \underline{\mathbb{C}}_{M_d}[\dim M_d]) \\
&= \bigoplus_{d \in \pi_1(G)} \xi^d \, H^\bullet(M_d, \Omega_{M_d}^\bullet)[\dim M_d] \\
&= \bigoplus_{d \in \pi_1(G)} \xi^d \, H_{\mathrm{dR}}^\bullet(M_d, \mathbb{C})[\dim M_d].
\end{aligned}
\tag{80}
$$

The cohomological grading is now manifest, while the secondary grading comes from the Hodge decomposition of de Rham cohomology. To be specific, a $(p, q)$-form cohomology class has cohomological and secondary degree

$$F = p + q - \dim M_d, \qquad R = p - \frac{1}{2}\dim M_d. \tag{81}$$

This coincides precisely with the space of supersymmetric ground states of a smooth $\mathcal{N} = (2, 2)$ sigma model [29] and so our proposal passes a consistency check. The general proposal (76) is a natural extension of this result to singular targets.

Finally, it is necessary to define a new R-symmetry $\widetilde{R}_C = R_C - \lambda \cdot J_C$ as in section 2.2 so that monopole operators are compatible with the fermion number $(-1)^F$, the cohomological and secondary gradings are shifted by an amount proportional to $d$,

$$\mathcal{H} = \sum_{d \in \pi_1(G)} (t^{-\frac{\lambda}{2}} \xi)^d \, \mathbb{H}^\bullet(M_d, P_d)[\lambda \cdot d]. \tag{82}$$

## 4.5 Mass Parameters

Let us now consider introducing real mass parameters $m$. If the moduli space $M_d$ is already compact and the supersymmetric quantum mechanics is gapped, introducing mass parameters does not change the supersymmetric ground states. More accurately, as explained in section 2.5, there is a flat Berry connection over the space of mass parameters. However, it is frequently the case that the moduli space $M_d$ is not compact. In this case, it is essential to introduce mass parameters and the definition of supersymmetric ground states (76) must be modified accordingly.

Introducing mass parameters $m$ modifies the vortex equations such that

$$(\sigma + m) \cdot X = 0, \qquad (\sigma + m) \cdot Y = 0, \tag{83}$$

where $\sigma$, $m$ are understood to act in the appropriate representations of $T$, $T_H$. This restricts solutions of the symplectic vortex equations invariant under the $U(1)_m \subset T_H$ generated by $m$. From the perspective of supersymmetric quantum mechanics, if $M_d$ is smooth, this introduces a perfect Morse-Bott function given by the moment map for $U(1)_m$. The supersymmetric ground states are then given by a Morse-Witten complex, which reduces to de Rham cohomology of the fixed locus.

We would now like to propose how this statement is generalised when $M_d$ is singular. For this purpose, we follow an algebraic perspective. The mass parameters will now generate a $\mathbb{C}_m^*$ action on $M_d$ with fixed locus

$$F_d = \bigsqcup_I F_{d,I}, \tag{84}$$

with disjoint components $F_{d,I}$ and corresponding attracting and repelling sets $M_{d,I}^{\pm}$. This decomposition depends only on the chamber $\mathfrak{c}_H$.

We now propose that the algebraic counterpart of introducing mass parameters in supersymmetric quantum mechanics is hyperbolic localisation [30]. In particular, there is a hyperbolic restriction functor $\Phi_d : D_c(M_d) \to D_c(F_d)$ for $\mathbb{C}_m^*$-equivariant constructible sheaves on $M_d$. The space of supersymmetric ground states can then be computed by hyperbolic localisation,

$$
\begin{aligned}
\mathcal{H} &= \sum_{d \in \pi_1(G)} \xi^d \, \mathbb{H}^\bullet(M_d, P_d) \\
&= \sum_{d \in \pi_1(G)} \xi^d \, \mathbb{H}^\bullet(F_d, \Phi(P_d)).
\end{aligned}
\tag{85}
$$

As mentioned above, if $M_d$ is not compact, the first line should be discarded and the second line considered a definition of the supersymmetric ground states. In this case, the space of supersymmetric ground states $\mathcal{H}$ will depend on the chamber $\mathfrak{c}_H$.

Let us check consistency with standard results in supersymmetric quantum mechanics and Morse theory. We consider again the simplest situation where $\mathfrak{M}_d = T^*[-1]M_d$ with $M_d$ a smooth projective variety and $P_d = \mathbb{C}_{M_d}[\dim M_d]$. In this instance, the hyperbolic restriction functor acts as follows,

$$
\Phi(\mathbb{C}_{M_d}) = \bigoplus_I \mathbb{C}_{F_{d,I}}[-\nu_{d,I}],
\tag{86}
$$

where

$$
\nu_{d,I} = \dim M_{d,I}^+ - \dim F_{d,I}.
\tag{87}
$$

Then

$$
\begin{aligned}
\mathbb{H}^\bullet(M_d, P_d) &= \bigoplus_I \mathbb{H}^\bullet(F_{d,I}, \underline{\mathbb{C}}_{F_{d,I}}[\dim M_d - \nu_{d,I}]) \\
&= \bigoplus_I H_{\mathrm{dR}}^\bullet(F_{d,I}, \mathbb{C})[\dim M_d - \nu_{d,I}].
\end{aligned}
\tag{88}
$$

Let us compare this result with an $\mathcal{N} = (2,2)$ supersymmetric quantum mechanics to $M_d$ with the perfect Morse-Bott function given by the moment map for $U(1)_m$. In this case, there are no instanton corrections and supersymmetric ground states coincide with the de Rham cohomology of the fixed locus with the degrees shifted by the Morse index $\nu_{d,I}$. In particular, for a $(p,q)$-form de Rham cohomology class on $F_{d,i}$, the corresponding supersymmetric ground state has

$$
F = p + q + \nu_{d,I} - \dim M_d, \qquad R = p + \frac{\nu_{d,I}}{2} - \frac{1}{2}\dim M_d.
\tag{89}
$$

The Morse indices $\nu_{d,i}$ for the moment map of a $U(1)_m$ action on a compact Kähler manifold $M_d$ coincide with the formula (87) and therefore we find a perfect match. The general proposal (85) is a natural generalisation to the case when $M_d$ is singular.

This construction will be exceptionally useful to allow explicit computation of supersymmetric ground states. One reason is that the fixed loci $F_{d,I}$ may be smooth, even when $M_d$ is not, allowing the computation to be reduced to the de Rham cohomology of $F_{d,I}$. Moreover, for theories satisfying the assumptions of section 3, the fixed components take the form of quasimaps to a point $I$, which can be rewritten as

$$
F_{d,I} = \bigsqcup_{\substack{\mathbf{d} \in \Lambda \\ |\mathbf{d}| = d}} \mathrm{Sym}^{\rho_{I,1}(\mathbf{d}) + (g-1)} \Sigma \times \cdots \times \mathrm{Sym}^{\rho_{I,k}(\mathbf{d}) + (g-1)} \Sigma.
\tag{90}
$$

Here $\Lambda$ is the co-character lattice of $G$, $|\cdot|$ is the projection of this lattice onto $\pi_1(G)$, whereas $\{\rho_{I,1}, \ldots, \rho_{I,k}\}$ are the weights in $\mathfrak{t}^*$ (of the G action on $T^*M$) selected at fixed point $I$. The de Rahm cohomology of this fixed locus is completely understood. This will enable a complete

determination of the space of supersymmetric ground states, which we build up in steps in section 5.

## 4.6 Recovering the Twisted Index

Let us now revisit the *A*-twisted index and provide a geometric interpretation of this observable using the above proposal.

First consider the limit of the twisted index defined in equation (39), which scales the parameters $x$ associated to the $T_H$ flavour symmetry in a way that depends on the chamber $\mathfrak{c}_H$. Recall that in this limit, the index only receives contributions from the supersymmetric ground states $\mathcal{H}$ in the chamber $\mathfrak{c}_H$. We find that

$$\lim_{\mathfrak{c}_H} \mathcal{I} = \sum_{d \in \pi_1(G)} \xi^d \, \widehat{\chi}_t(M_d, P_d), \tag{91}$$

where

$$\widehat{\chi}_t(M_d, P_d) := \sum_{p,q} (-1)^{p+q} t^p h^{p,q}(\mathbb{H}^\bullet(M_d, P_d)) \tag{92}$$

and $h^{p,q}$ denotes the dimensions of the graded components of the Hodge structure. When $M_d$ is a smooth projective variety and $P_d = \mathbb{C}_M[\dim M_d]$,

$$
\begin{aligned}
\widehat{\chi}_t(M_d, P_d) &= \sum_{p,q=1}^{\dim M_d} (-1)^{p+q} t^p \left( (-1)^{\dim M_d} t^{-\frac{\dim M_d}{2}} h^{p,q}(M_d) \right) \\
&= (-1)^{\dim M_d} t^{-\frac{\dim M_d}{2}} \sum_{p,q=1}^{\dim M_d} (-1)^{p+q} t^p h^{p,q}(M_d) \\
&= (-1)^{\dim M_d} t^{-\frac{\dim M_d}{2}} \chi_{-t}(M_d) \\
&=: \widehat{\chi}_t(M_d),
\end{aligned}
\tag{93}
$$

which is a symmetrised version of the standard Hirzebruch genus of $M_d$. The index in (92) is then a natural generalisation to singular $M_d$.

Finally, in the limit $t \to 1$ the result is automatically independent of the parameters $x$ associated to the flavour symmetry $T_H$ without taking a further limit, and the result reproduces the generalised Euler number

$$\lim_{t \to 1} \mathcal{I} = \sum_{d \in \pi_1(G)} \xi^d \, \widehat{e}(M_d, P_d), \tag{94}$$

where

$$\widehat{e}(M_d, P_d) = \sum_{p,q=1}^{\dim M_d} (-1)^{p+q} h^{p,q}(\mathbb{H}^\bullet(M_d, P_d)). \tag{95}$$

When $M_d$ is a smooth projective variety and $P_d = \mathbb{C}_M[\dim M_d]$,

$$
\begin{aligned}
\widehat{e}(M_d, P_d) &= (-1)^{\dim M_d} \sum_{p,q}^{\dim M_d} (-1)^{p+q} h^{p,q}(M_d) \\
&= (-1)^{\dim M_d} e(M_d) \\
&=: \widehat{e}(M_d),
\end{aligned}
\tag{96}
$$

which is a shifted version of the standard Euler number.

These limits of the index further decompose as sum of generalised Hirzebruch genera of Euler numbers of the fixed loci $F_{d,I}$. In view of (90), this will turn out to be exceptionally powerful when the assumptions spelled out in 3.1 are imposed. We illustrate this in examples in section 5.

# 5 A-twist examples

In this section, we compute the space of supersymmetric ground states in the *A*-twist in a series of examples, building up to a general result for theories satisfying the assumptions of section 3.

## 5.1 Hypermultiplet

Although outside of the class of theories defined in section 3.1, it is convenient to investigate the free hypermultiplet. This will introduce the charge assignments and geometric interpretation of supercharges that will become crucial later on. The free hypermultiplet has flavour symmetry $T_H \cong U(1)$ and a real mass parameter $m \in \mathbb{R}$ with two chambers: $\mathfrak{c}_H^{\pm} = \{\pm m > 0\}$.

To compute supersymmetric ground states, it is convenient to introduce an extra ingredient: a background holomorphic line bundle $L$ of degree $d$ on $\Sigma$ associated to the symmetry. This is compatible with the four supercharges preserved in the *A*-twist. Let us fix a choice of spin structure $S = K_\Sigma^{1/2}$ and define the numbers

$$n_X := h^0(\Sigma, K_\Sigma^{1/2} \otimes L), \qquad n_Y := h^0(\Sigma, K_\Sigma^{1/2} \otimes L^{-1}), \tag{97}$$

where $n_X - n_Y = d$ by Serre duality.

The effective $\mathcal{N} = (2,2)$ supersymmetric quantum mechanics consists of $n_X + n_Y$ free chiral multiplets. Keeping track of the secondary grading and global symmetry, this can be regarded as a sigma model with target

$$M = x t^{-\frac{1}{2}} \mathbb{C}^{n_X} \oplus x^{-1} t^{-\frac{1}{2}} \mathbb{C}^{n_Y}. \tag{98}$$

From the perspective of the subalgebra generated by $Q_+$, each chiral multiplet decompose into an $\mathcal{N} = (0,2)$ chiral multiplet and an $\mathcal{N} = (0,2)$ Fermi multiplet, which we can regard as an $\mathcal{N} = (0,2)$ quantum mechanics with target $\mathfrak{M} = T^*[-1]M$.

Supersymmetric ground states are $L^2$-harmonic $(p,q)$-forms on $M$. Due to the non-compactness, the quantum mechanics is not gapped as it stands. This is cured by introducing a real mass parameter, which deforms the supercharges to

$$Q_+ = e^{-h} \bar{\partial} e^h, \qquad Q_- = e^{-h} \partial e^h, \tag{99}$$

where $h$ is the moment map for the $U(1)_m$ action on $M$. Let us introduce coordinates $x_a, y_{a'}$ on $M$ with $a = 1, \ldots, n_X$ and $a' = 1, \ldots, n_Y$. Then

$$h = m \left( \sum_{a=1}^{n_X} |x_a|^2 - \sum_{a'=1}^{n_Y} |y_{a'}|^2 \right). \tag{100}$$

There is now a single supersymmetric ground state in each chamber, with Gaussian wavefunction of the form

$$
\begin{aligned}
e^{-m\left(\sum_{a=1}^{n_X} |x_a|^2 + \sum_{a'=1}^{n_Y} |y_{a'}|^2\right)} \prod_{a'=1}^{n_Y} dy_{a'} \wedge d\bar{y}_{a'}, \qquad & m > 0, \\
e^{+m\left(\sum_{a=1}^{n_X} |x_a|^2 + \sum_{a'=1}^{n_Y} |y_{a'}|^2\right)} \prod_{a=1}^{n_X} dx_a \wedge d\bar{x}_a, \qquad & m < 0.
\end{aligned}
\tag{101}
$$

These are harmonic representatives of $L^2$-cohomology classes on $M$ with respect to the deformed de Rham supercharge $Q = e^{-h} d e^h$.

The cohomological grading is given by

$$F = \nu - \frac{1}{2} \dim_{\mathbb{R}} M, \tag{102}$$

where

$$
\nu = p + q = \begin{cases} 2n_Y & m > 0 \\ 2n_X & m < 0 \end{cases}, \tag{103}
$$

is the real Morse index of $h$ at the origin. The supersymmetric ground state therefore has fermion number $F = \mp d$ when $\pm m > 0$. The secondary grading is given by

$$
R = p - \frac{1}{2}\dim_{\mathbb{C}} M \tag{104}
$$

and therefore the supersymmetric ground state has $R = \mp d/2$ when $\pm m > 0$. Finally, as expected on general grounds, the supersymmetric ground state is uncharged under the global symmetry. In summary, using the notation introduced in (17)

$$
\mathcal{H} = \begin{cases} t^{-\frac{d}{2}} \mathbb{C}[+d] & m > 0 \\ t^{+\frac{d}{2}} \mathbb{C}[-d] & m < 0 \end{cases}. \tag{105}
$$

Note that this result depends on the holomorphic line bundle $L$ only through its degree $d$, and does not depend on the choice of spin structure $K_{\Sigma}^{1/2}$.

Finally, let us check compatibility with the twisted index. The general $A$-twisted index counting states in the cohomology of $Q_+$ is in our conventions

$$
\mathcal{I} = \mathrm{Tr}_{\mathcal{H}^{1/2}}(-1)^F t^R x^{J_H} = \left( \frac{x - t^{-\frac{1}{2}}}{1 - x t^{-\frac{1}{2}}} \right)^d. \tag{106}
$$

The contribution of supersymmetric ground states to the twisted can be extracted by sending $|m| \to \infty$ in the appropriate chamber. This corresponds to taking the limit $x^{\pm 1} \to 0$ in the chamber $\mathfrak{c}_H^{\pm} = \{\pm m > 0\}$. The result is

$$
\lim_{x^{\pm 1} \to 0} \mathcal{I} = (-1)^d t^{\mp \frac{d}{2}} \tag{107}
$$

in complete agreement with our computation of the supersymmetric ground states. Alternatively, the limit guaranteed to count supersymmetric ground states is

$$
\lim_{t \to 1} \mathcal{I} = (-1)^d, \tag{108}
$$

providing a slightly weaker check.

## 5.2 SQED, 1 hypermultiplet

Now consider $G = U(1)$ with one hypermultiplet of charge $+1$. There is now a topological flavour symmetry $T_C \cong U(1)$ with real FI parameter $\zeta$ and two chambers $\mathfrak{c}_C = \{\pm \zeta > 0\}$.

Provided the normalised FI parameter is such that $\widetilde{\zeta} \notin \mathbb{Z}$, the system localises onto solutions of the symplectic vortex equations (60), which become

$$
\frac{1}{e^2} * F_A + |X|^2 - |Y|^2 = \zeta, \tag{109}
$$

$$
XY = 0, \qquad \bar{\partial}_A X = 0, \qquad \bar{\partial}_A Y = 0, \tag{110}
$$

where

$$
X \in \Gamma(\Sigma, K_{\Sigma}^{1/2} \otimes P), \qquad Y \in \Gamma(\Sigma, K_{\Sigma}^{1/2} \otimes P^{-1}) \tag{111}
$$

and $P$ denotes the principle $U(1)$ bundle on $\Sigma$. We are interested in the moduli space of solutions to these equations in the infinite-tension limit $e^2 \mathrm{vol}(\Sigma) \to \infty$ (or equivalently $|\widetilde{\zeta}| \to \infty$) with fixed FI parameter $\zeta$ in a given chamber.

The moduli space is a disjoint union of components labelled by the topological degree $d \in \mathbb{Z}$ of $P$. Each component $M_d$ is the moduli space of solutions to the abelian vortex equations,

$$\zeta > 0 \quad : \quad \frac{1}{e^2} * F_A + |X|^2 = \zeta, \qquad \bar{\partial}_A X = 0, \qquad Y = 0,$$
$$\zeta < 0 \quad : \quad \frac{1}{e^2} * F_A - |Y|^2 = \zeta, \qquad \bar{\partial}_A Y = 0, \qquad X = 0, \tag{112}$$

which is a smooth compact Kähler manifold. This has a standard algebraic description as a symmetric product

$$M_d = \mathrm{Sym}^n \Sigma, \tag{113}$$

with

$$n = \begin{cases} +d + g - 1 & \zeta > 0 \\ -d + g - 1 & \zeta < 0 \end{cases}. \tag{114}$$

The symmetric product parametrises a holomorphic line bundle $L$ of degree $d$, together with a non-vanishing holomorphic section of $K_\Sigma^{1/2} \otimes L$ when $\zeta > 0$ or $K_\Sigma^{1/2} \otimes L^{-1}$ if $\zeta < 0$. If $n < 0$ then that component of the moduli space is empty. In this example $M_d$ is smooth and the derived moduli space is simply $\mathfrak{M}_d = T^*[-1]M_d$.

The effective supersymmetric quantum mechanics is a smooth $\mathcal{N} = (2, 2)$ sigma model with target $M_d$. Since the target is compact Kähler, supersymmetric ground states are harmonic $(p, q)$-forms on $M_d$, or equivalently de Rham cohomology supplemented with the Hodge decomposition.

Ordinarily, supersymmetric ground states coming from $(p, q)$-forms on a space of complex dimension $d$ would have cohomological and secondary grading

$$F = p + q - n, \qquad R_+ = p - \frac{n}{2} \tag{115}$$

under the identifications $F = R_C$ and $R_+ = \frac{1}{2}(R_C - R_H)$. However, in this instance this assignment is incompatible with $(-1)^F$ as the fermion number because bosonic monopole operators $u^\pm$ in the three-dimensional theory would have $F = 1$.

The solution is to define a new R-symmetry

$$\widetilde{R}_C = R_C - J_C \tag{116}$$

and instead identify $F = \widetilde{R}_C$ and $R_+ = \frac{1}{2}(\widetilde{R}_C - R_H)$. This ensures that the monopole operators $u^+, u^-$ have even fermion number $F = 0, 2$, while leaving the assignments of the elementary fields unchanged. This modifies the grading of supersymmetric ground states such that a $(p, q)$-form cohomology class has weight

$$F = p + q - \widetilde{n}, \qquad R_+ = p - \frac{\widetilde{n}}{2}, \tag{117}$$

where $\widetilde{n} := n \pm d$ in the chambers $\pm\zeta > 0$, since supersymmetric ground states arising from cohomology classes on $M_d$ carry topological charge $\mp d$.

The space of supersymmetric ground states therefore consist of all $(p, q)$-form cohomology classes of symmetric products $\mathrm{Sym}^n \Sigma$ for $n \geq 0$, with the cohomological and secondary gradings determined as in the previous paragraph. The cohomology of symmetric products is well understood from [31] and summarised in appendix A. From equation (196), we find

$$\mathcal{H} = \begin{cases} (\xi t^{-\frac{1}{2}})^{1-g} (\mathrm{Sym}^\bullet V)[1-g] & \zeta > 0 \\ (\xi t^{-\frac{1}{2}})^{g-1} (\mathrm{Sym}^\bullet V^\vee)[g-1] & \zeta < 0 \end{cases}$$
$$= \begin{cases} \widehat{\mathrm{Sym}}^\bullet V & \zeta > 0 \\ \widehat{\mathrm{Sym}}^\bullet V^\vee & \zeta < 0 \end{cases}, \tag{118}$$

where

$$V = \xi\left(\mathbb{C} \oplus t^{-1}\mathbb{C}^g[1] \oplus \mathbb{C}^g[1] \oplus t^{-1}\mathbb{C}[2]\right) \tag{119}$$

and

$$\widehat{\mathrm{Sym}}^\bullet V := (\det V)^{\frac{1}{2}} \cdot \mathrm{Sym}^\bullet V. \tag{120}$$

Note that the space of supersymmetric ground states is manifestly a Fock space. The generators can be thought of as the descendants of monopole operators $u^+$ (when $\zeta > 0$) and $u^-$ (when $\zeta < 0$) integrated over homology classes on $\Sigma$.

Computing the trace over supersymmetric ground states in either chamber, we find

$$\mathrm{Tr}_{\mathcal{H}}(-1)^F t^R \xi^{J_C} = \begin{cases} \sum_{d\geq 0}(-t^{-\frac{1}{2}}\xi)^{d-g+1}\widehat{\chi}_t(\mathrm{Sym}^d\Sigma) & \zeta > 0 \\ \sum_{d>0}(-t^{\frac{1}{2}}\xi^{-1})^{d-g+1}\widehat{\chi}_t(\mathrm{Sym}^d\Sigma) & \zeta < 0 \end{cases} \tag{121}$$

$$= (-t^{-\frac{1}{2}}\xi)^{1-g}(1-\xi)^{g-1}(1-t^{-1}\xi)^{g-1},$$

where

$$\widehat{\chi}_t(M) := (-t^{-\frac{1}{2}})^{\dim M}\chi_{-t}(M) \tag{122}$$

is the normalised Hirzebruch genus. Despite the existence of supersymmetric ground states with arbitrarily large topological charge, the trace is a finite Laurent polynomial due to

$$\widehat{\chi}_t(\mathrm{Sym}^d\Sigma) = 0 \quad \text{if} \quad d > 2g-2. \tag{123}$$

This is a consequence of the fact that the symmetric product $\mathrm{Sym}^d\Sigma$ is a smooth fibre bundle over the torus $\mathrm{Pic}^d(\Sigma) \cong T^{2g}$ when $d > 2g-2$.

Since symmetric products are smooth compact Kähler, the Hodge to de Rham spectral sequence collapses and $\mathcal{H}^{1/2} \cong \mathcal{H}$. The above expression should therefore coincide with the full twisted index $\mathcal{I}$. Indeed, taking into account the shifted $R$-symmetry $\widetilde{R}_C$, the contour integral for the twisted index is

$$\mathcal{I} = -(t^{\frac{1}{2}}-t^{-\frac{1}{2}})^{g-1}\sum_{d\in\mathbb{Z}}(-t^{-\frac{1}{2}}\xi)^d \int_\Gamma \frac{dz}{2\pi i z}\left(\frac{z-t^{-\frac{1}{2}}}{1-zt^{-\frac{1}{2}}}\right)^d \left[\frac{(1-t^{-1})z}{(1-zt^{-\frac{1}{2}})(z-t^{-\frac{1}{2}})}\right]^g, \tag{124}$$

where $\Gamma$ evaluates the residue at $zt^{-\frac{1}{2}} = 1$ when $\zeta > 0$ and minus the residue at $zt^{\frac{1}{2}} = 1$ when $\zeta < 0$. The sum of residues at $z = 0$ and $z = \infty$ vanishes and this reproduces the above result as an expansion in either chamber.

## 5.3 SQED, $N$ hypermultiplets

We now extend this computation to $G = U(1)$ and $N$ hypermultiplets of charge $+1$. The topological symmetry is $T_C \cong U(1)$ and we introduce a real FI parameter $\zeta$ with two chambers $\mathfrak{c}_C = \{\zeta > 0\}$ and $\{\zeta < 0\}$. There is now a flavour symmetry $T_H = U(1)^N/U(1)$ transforming the hypermultiplets and real mass parameters $(m_1,\ldots,m_N)$ satisfying $\sum_j m_j = 0$, with $N!$ possible chambers labelled by the ordering of distinct parameters. Our default chambers are $\mathfrak{c}_C = \{\zeta > 0\}$ and $\mathfrak{c}_H = \{m_1 > m_2 > \cdots > m_N\}$.

Provided the normalised FI parameter is such that $\widetilde{\zeta} \notin \mathbb{Z}$, and setting the mass parameters to zero, the system localises onto solutions of the symplectic vortex equations (60), which become

$$\frac{1}{e^2} * F_A + \sum_j |X_j|^2 - |Y_j|^2 = \zeta, \tag{125}$$

$$\sum_j X_j Y_j = 0, \qquad \bar{\partial}_A X_j = 0, \qquad \bar{\partial}_A Y_j = 0. \tag{126}$$

The moduli space $\mathfrak{M}$ is a disjoint union of components $\mathfrak{M}_d$ labelled by the topological degree $d \in \mathbb{Z}$. Each component has an algebraic description parametrising:

- a holomorphic line bundle $L$ of degree $d$,

- holomorphic sections $X_j \in H^0(K_\Sigma^{1/2} \otimes L)$ and $Y_j \in H^0(K_\Sigma^{1/2} \otimes L^{-1})$ subject to the constraint $\sum_j X_j Y_j = 0$ and a stability condition depending on $\zeta$.

This is the moduli space of $\widetilde{\zeta}$-stable twisted quasi-maps $\Sigma \to T^* \mathbb{CP}^{N-1}$ of degree $d$.

In general, the moduli space has a complicated dependence on the parameters $g$, $\zeta$ and $d$. As usual, we consider the infinite-tension limit $|\widetilde{\zeta}| \to \infty$ with $\zeta$ fixed. To streamline the presentation, we will present intermediate steps in the chamber $\mathfrak{c}_C = \{\zeta > 0\}$. There are then three distinct regions for the degree $d$, which we analyse in turn. We summarise a uniform result for the supersymmetric ground states at the end.

**Region I:** $d < g - 1$

When $d < g - 1$, a vanishing theorem ensures $X_j = 0$ for all $j = 1, \ldots, N$. This is incompatible with equation (125) when $\zeta > 0$ and therefore $\mathfrak{M}_d = \emptyset$. There are therefore no supersymmetric ground states with $d < g - 1$ in the chamber $\zeta > 0$.

**Region II:** $d > g - 1$

In the opposite region, $d > g - 1$, a vanishing theorem ensures $Y_j = 0$ for all $j = 1, \ldots, N$. This is leads to a dramatic simplification such that $\mathfrak{M}_d = T^*[-1]M_d$ where $M_d$ parametrises solutions to the abelian vortex equations

$$\frac{1}{e^2} * F_A + \sum_{j=1}^N |X_j|^2 = \zeta, \qquad \bar{\partial}_A X_i = 0. \tag{127}$$

This is a smooth compact Kähler manifold of complex dimension

$$n = Nd + g - 1, \tag{128}$$

which is a fiber bundle over $\text{Pic}^d(\Sigma) \cong T^{2g}$ with fiber $\mathbb{CP}^{Nd-1}$. From an algebraic perspective, it is a projective variety parametrising twisted quasi-maps $\Sigma \to \mathbb{CP}^{N-1}$ of degree $d$, or equivalently a holomorphic line bundle $L$ of degree $d$, together with $N$ holomorphic sections of $K_\Sigma^{1/2} \otimes L$ that do not simultaneously vanish.

The effective quantum mechanics is therefore a smooth $\mathcal{N} = (2,2)$ sigma model and supersymmetric ground states coincide with de Rham cohomology of $M_d$. It is again necessary to define a new R-symmetry $\widetilde{R}_C = R_C - NJ_C$ such that the monopole operators $u^+$, $u^-$ have even fermion number $F = 0, 2N$. A supersymmetric ground state coming from a $(p,q)$-form cohomology class on $M_d$ then has weights

$$F = p + q - \widetilde{n}, \qquad R_+ = p - \frac{\widetilde{n}}{2}, \tag{129}$$

where $\widetilde{n} := n + Nd$.

To compute the ground states explicitly we turn on real masses $m = (m_1, \ldots, m_N)$. This introduces a real superpotential given by the moment map for the $U(1)_m \subset T_H$ action on $M_d$ generated by the mass parameters. Provided the masses are generic, meaning $m_i \neq m_j$ for $i \neq j$, critical points on $X = \mathbb{CP}^{N-1}$ correspond to choices $I = \{i\}$ of fields $X_i$ that have a non-zero VEV, all the others being set to zero. The $I$-th component of the critical locus parametrises

solutions of the abelian vortex equations where $X_j = 0$ for $j \neq i$. This corresponds to quasi-maps $\Sigma \to p_I \subset \mathbb{CP}^{N-1}$. Thus

$$(M_d)^{T_H} = \bigsqcup_{I=1}^{N} \mathrm{Sym}^{n_I}\Sigma, \tag{130}$$

where

$$n_I = d + g - 1. \tag{131}$$

Following the standard procedure in supersymmetric quantum mechanics, we construct perturbative ground states from cohomology classes on the critical locus. The gradings of these states depend on the Morse index of the components of the critical locus. Let us fix the chamber $\mathfrak{c}_H = \{m_1 > m_2 > \cdots > m_N\}$. Then the real Morse index of the $I$-th fixed component (corresponding to the field $X_i$) is

$$\nu_I = 2(N-i)d \tag{132}$$

and a perturbative ground state arising from a $(p,q)$-form on the $I$-th component is

$$F = p + q - \widetilde{n}_I, \qquad R_+ = p - \frac{\widetilde{n}_I}{2}, \tag{133}$$

where

$$\widetilde{n}_I = n_I - \nu_I. \tag{134}$$

Since the Morse function $h$ is the moment map for a hamiltonian $U(1)$ isometry of a smooth Kähler manifold, the Morse indices are even and there are no instanton corrections. So these are honest supersymmetric ground states.

**Region III:** $|d| < g - 1$

In this region, $M_d$ is singular and $\mathfrak{M}_d$ is not a shifted cotangent bundle. The supersymmetric ground states should then be computed from the hyper-cohomology

$$\mathbb{H}^\bullet(M_d, P_d), \tag{135}$$

where $P_d$ is the perverse sheaf induced by the shifted symplectic structure on $\mathfrak{M}_d$.

However, turning on generic mass parameters $(m_1, \ldots, m_N)$ restricts to configurations where either $X_j = 0$ or $Y_j = 0$ individually for each $j = 1, \ldots, N$. This fixed locus is therefore a disjoint union of smooth components parametrising abelian vortices,

$$F_d = \bigsqcup_{I=1}^{N} F_{d,I}, \qquad F_{d,I} = \mathrm{Sym}^{n_I}\Sigma. \tag{136}$$

Our general proposal for the supersymmetric ground states is then the hyper-cohomology of $\Phi_d(P_d)$, where $\Phi_d : D_c(M_d) \to D_c(F_d)$ is the hyperbolic restriction functor. The fixed locus $F_d$ is smooth and lies away from singularities in $M_d$, such that the normal bundle is identical to region II. We then expect that

$$\Phi(\mathbb{C}_{M_d}) = \bigoplus_I \mathbb{C}_{F_{d,I}}[-\nu_{d,I}], \tag{137}$$

where $\nu_{d,I}$ coincide with the Morse indices (132). We therefore propose that the result for supersymmetric ground states from region II is extrapolated without change to $d \geq 1 - g$.

**Summary**

We can now re-cycle our result from the computations of supersymmetric ground states from the de Rham cohomology of symmetric products in the case $N = 1$, with degree shifts. The result for the space of supersymmetric ground states in the chambers $\mathfrak{c}_C = \{\zeta > 0\}$ and $\mathfrak{c}_H = \{m_1 > m_2 > \cdots > m_N\}$ is given by

$$
\begin{aligned}
\mathcal{H} &= \bigoplus_{I=1}^{N} (\xi t^{\frac{1}{2}-i})^{1-g} (\mathrm{Sym}^\bullet V_I)[(1-g)(2i-1)] \\
&= \bigoplus_{I=1}^{N} \widehat{\mathrm{Sym}}^\bullet V_I \,,
\end{aligned}
\tag{138}
$$

where

$$
V_I = \xi t^{-i} (\mathbb{C} \oplus \mathbb{C}^g[-1] \oplus t\,\mathbb{C}^g[-1] \oplus t\,\mathbb{C}[-2])[2i] \,.
\tag{139}
$$

The results in other chambers can be obtained by conjugating $V_I$ and permuting their assignment to fixed points.

The character of the space of supersymmetric ground states is

$$
\mathrm{Tr}_{\mathcal{H}} (-1)^F t^{R_+} \xi^{J_C} = \sum_{j=1}^{N} (-t^{j-\frac{1}{2}} \xi)^{1-g} (1 - t^{1-j}\xi)^{g-1} (1 - t^{-j}\xi)^{g-1} \,.
\tag{140}
$$

In the limit $t \to 1$, the contribution from each fixed point becomes identical and

$$
\mathrm{Tr}_{\mathcal{H}} (-1)^F \xi^{J_C} = N(-\xi)^{1-g} (1-\xi)^{2g-2} \,.
\tag{141}
$$

Let us check compatibility of these results with the *A*-twisted index, which has the contour integral representation

$$
\mathcal{I} = -(t^{-\frac{1}{2}} - t^{\frac{1}{2}})^{g-1} \sum_{d \in \mathbb{Z}} ((-1)^N t^{-\frac{N}{2}} \xi)^d \int_\Gamma \frac{dz}{2\pi i z} \prod_{j=1}^{N} \left( \frac{z x_j - t^{-\frac{1}{2}}}{1 - z x_j t^{-\frac{1}{2}}} \right)^d H^g \,,
\tag{142}
$$

where $H$ is the Hessian. Here $x_i = e^{-\beta(m_i + i a_C)}$ and $\Gamma$ evaluates the residues at $z t^{-1/2} x_j = 1$ in the chamber $\zeta > 0$ and minus the residues at $z x_j t^{1/2} = 1$ in the chamber $\zeta < 0$. In the limit $t \to 1$ that is guaranteed to receive contributions from supersymmetric ground states only, it is straightforward to check that

$$
\lim_{t \to 1} \mathcal{I} = N(-\xi)^{1-g} (1-\xi)^{2g-2}
\tag{143}
$$

in perfect agreement with our construction of supersymmetric ground states. We can make a more detailed comparison by keeping $t$ but scaling the masses to manually project onto supersymmetric ground states. We do this by sending the mass parameters to infinity in the chamber $\mathfrak{c}_H = \{m_1 > m_2 > \cdots > m_N\}$. The result is

$$
\lim_{|m| \to \infty} \mathcal{I} = \sum_{j=1}^{N} (-t^{j-\frac{1}{2}} \xi)^{1-g} (1 - t^{1-j}\xi)^{g-1} (1 - t^{-j}\xi)^{g-1} \,,
\tag{144}
$$

in agreement with the supersymmetric ground states.

### 5.4 General class

We can now readily generalise the discussion and compute the space of supersymmetric ground states of theories satisfying Assumptions I and II of section 3.1. We fix an FI parameter $\zeta \in \mathfrak{c}_C$ and associated Higgs branch $X$.

First, recall that fixed points $I$ on $X$ are labelled by collections of weights $\{\rho_1, \dots \rho_k\} \in \mathfrak{t}^*$ specifying the non-vanishing hypermultiplet fields. They satisfy conditions summarised in section 3.2. The components of the fixed locus of the moduli space $M_d$ are labelled in the same way and take the form of symmetric products

$$F_{d,I} = \bigsqcup_{\substack{\mathbf{d} \in \Lambda \\ |\mathbf{d}| = d}} \prod_{a=1}^{k} \mathrm{Sym}^{\rho_{I,a}(\mathbf{d}) + (g-1)} \Sigma \,. \tag{145}$$

Here $\mathbf{d}$ is a GNO flux valued in the co-character lattice $\Lambda$ of $G$, and $|\cdot|$ denotes the projection of this lattice onto $\pi_1(G)$. The symmetric products parametrize the zeros of the sections corresponding to these weights. The fixed components $F_{d,I}$ correspond to twisted quasi-maps of degree $d$ to a fixed point $p_I$ on $X$.

Under our assumptions, we can reduce the problem to multiple copies of supersymmetric QED with one hypermultiplet. The idea is to disentangle the powers of the symmetric products in (145) by setting

$$(\rho_{I,1}(\mathbf{d}) + (g-1), \cdots, \rho_{I,k}(\mathbf{d}) + (g-1)) := (n_1, \cdots, n_k) \tag{146}$$

and then make use of (118).

In practice, the procedure works as follows. Let $\rho_I$ be the $k \times k$ matrix whose rows are the weights $\{\rho_{I,1}, \dots \rho_{I,k}\}$ associated to the fixed point $I$. Since the weights $\{\rho_{I,1}, \dots \rho_{I,k}\}$ are linearly independent, we define the inverse matrix

$$\omega_I := \rho_I^{-1} \,. \tag{147}$$

We now define $\omega_{I,a}$ to be the $a$-th row of the inverse matrix, which is an element of the co-character lattice $\Lambda$, and form the monomial

$$\xi^{I,a} := \xi^{|\omega_{I,a}|} \,. \tag{148}$$

Let $\Lambda^{\pm} \subset \Lambda$ correspond to those elements of the co-character lattice whose pairing with the FI parameter $\zeta$ is positive and negative respectively. We can collect

$$\begin{aligned}
C_I^+ &:= \bigoplus_{\omega_{I,a} \in \Lambda^+} \xi^{I,a} \mathbb{C} \oplus \bigoplus_{\omega_{I,a} \in \Lambda^-} \xi^{-I,a} t^{-1} \mathbb{C}[2], \\
C_I^- &:= \bigoplus_{\omega_{I,a} \in \Lambda^-} \xi^{I,a} \mathbb{C} \oplus \bigoplus_{\omega_{I,a} \in \Lambda^+} \xi^{-I,a} t \mathbb{C}[-2]
\end{aligned} \tag{149}$$

and set

$$V_I := t^{-\tilde{d}_I/2} \left( C_I^+ \oplus \left( C_I^- \right)^\vee \oplus \left( C_I^+[1] \oplus \left( C_I^-[1] \right)^\vee \right) \otimes \mathbb{C}^g \right) [\tilde{d}_I] \,. \tag{150}$$

Here $\tilde{d}_I$ are shifts that depend on the Morse index of the fixed locus. It is easily checked that the above expression encodes the expansion of the cohomologies of $k$ products of symmetric products. We have arranged terms in a slightly counter-intuitive way so that comparison with the B-twist will be immediate. Then in view of (118) we have

$$\mathcal{H} = \bigoplus_I \widehat{\mathrm{Sym}}^\bullet V_I \,. \tag{151}$$

Based on this expression it is straighforward to compute the index. For simplicity, we do this in the $t \to 1$ limit. Using the formula (198)

$$\sum_{d \in \mathbb{N}} x^d \left( \sum_{k=0}^{d} (-1)^k H_{dR}^k \left( \text{Sym}^d(\Sigma) \right) \right) = (1-x)^{2(g-1)}, \tag{152}$$

and we find

$$\lim_{t \to 1} \mathcal{I} = \sum_I \prod_{a=1}^{k} (-1)^{(1-g)\widetilde{d}_I} \left( \frac{\xi^{I,a/2}}{1 - \xi^{I,a}} \right)^{2-2g}. \tag{153}$$

The resummation is possible because the selected weights satisfy the JK condition (51).

## 6 Localisation in the $B$-Twist

### 6.1 Decomposing Supermultiplets

In the $B$-twist, 3d $\mathcal{N} = 4$ supermultiplets decompose into 1d $\mathcal{N} = (0,4)$ supermultiplets. A supersymmetric gauge theory can be regarded as an infinite-dimensional gauged supersymmetric quantum mechanics as follows. As before, let $P$ denote a principle $G$-bundle on $\Sigma$ with connection $A$. We have the following supermultiplets:

- A 1d $\mathcal{N} = (0,4)$ vectormultiplet for the infinite-dimensional gauge group of automorphisms of $P$. The bosonic components are $A_3$, $\sigma$, and auxiliary fields $D_{1d}^{AB}$.

- A 1d $\mathcal{N} = (0,4)$ twisted hypermultiplet $(\bar{\partial}_A, \varphi)$ with $\varphi$ transforming as a section of $\Omega^1(P_{\mathfrak{g}})$.

- A 1d $\mathcal{N} = (0,4)$ hypermultiplet $(X, Y)$ transforming as sections of $(P \times_G T^*M)$.

- A 1d $\mathcal{N} = (0,4)$ Fermi multiplet transforming as sections of $(P \times_G T^*M)$.

The supermultiplets are accompanied by superpotential couplings from an $\mathcal{N} = (0,2)$ perspective that are necessary to ensure enhancement to $\mathcal{N} = (0,4)$ supersymmetry. The relevant superpotentials are very similar to those that arise in the decomposition of 2d $\mathcal{N} = (4,4)$ supermultiplets into 2d $\mathcal{N} = (0,4)$ supersymmetry [32].

### 6.2 Localising to the Higgs Branch

For localisation, it is again convenient to use 1d $\mathcal{N} = (0,4)$ Lagrangians for the supermultiplets introduced above. We introduce exact Lagrangians $e^{-2}L_V$, $e^{-2}L_{tH}$, $L_H$, $L_F$ for the vectormultiplet, twisted hypermultiplet, hypermultiplet and Fermi multiplet respectively.

We will again need to decompose the 3d FI parameter Lagrangian

$$\begin{aligned} L_{\text{FI}} &= \frac{i\zeta}{2\pi} D_{12} \\ &= \frac{i\zeta}{2\pi} D_{1d,12} + \frac{\zeta}{2\pi} \left( *F_A + 2[\varphi, \varphi^\dagger] \right). \end{aligned} \tag{154}$$

The remaining Lagrangians $L_m$ and $L_\zeta$ are the contributions from mass and 3d FI parameters and are not exact.

Our starting point for supersymmetric localisation will be the Lagrangian

$$L = \frac{1}{t^2} \left( \frac{1}{e^2} (L_V + L_{tH}) + L_H + L_{1d\text{FI}} \right) + L_\zeta + L_m, \tag{155}$$

where we have introduced a positive constant $t^2$ in front of a particular combination of exact Lagrangians. Let us first set the mass parameters to vanish, $m = 0$. In the limit $t^2 \to 0$, the supersymmetric quantum mechanics localises onto solutions of

$$
\mu^{AB} = \zeta^{AB}, \qquad [\sigma, \varphi] = 0, \qquad d_A \sigma = 0,
$$
$$
\frac{1}{e^2} * F_A + 2[\varphi, \varphi^\dagger] = 0, \qquad \bar{\partial}_A \varphi = 0,
$$
$$
\sigma \cdot X^A = 0, \qquad \varphi \cdot X^A = 0, \qquad \sigma \cdot X^{\dagger A} = 0, \qquad \varphi \cdot X^{\dagger A} = 0,
$$
$$
\bar{\partial}_A X^A = 0, \qquad \bar{\partial}_A X^{\dagger A} = 0. \tag{156}
$$

We have used $SU(2)_H$ covariant notation such that the hypermultiplet components are $X^A = (X, Y^\dagger)$ and $X^{\dagger A} = (Y, -X^\dagger)$. This symmetry is broken to a maximal torus $U(1)_H$ by a non-vanishing FI parameter $\zeta := \zeta^{12}$ but it is nevertheless convenient to manifest this structure in the equations.

The moduli space of solutions to this system of equations has an intricate structure depending on the data $(G, R)$. This can involve hypermultiplet branches parametrised by $(X, Y)$, twisted hypermultiplet branches parametrised by $(\bar{\partial}_A, \varphi)$ and various mixed branches. There can also be branches with continuous unbroken gauge symmetry parameterised by the vector-multiplet scalar $\sigma$.

In spite of this complexity, there are two important features that can be noticed. First, the 1d $\mathcal{N} = (0, 4)$ twisted hypermultiplet $(\bar{\partial}_A, \varphi)$ obeys Hitchin's equations. In particular, the real equation implies that the degree vanishes, $d = 0$. Second, the 1d $\mathcal{N} = (0, 4)$ hypermultiplet is now covariantly constant, $d_A X = d_A Y = 0$, and obey the same moment map constraints defining the Higgs branch $X$.

It is convenient to pass to an algebraic description where $\bar{\partial}_A$ parametrises the complex structure on a complex vector bundle $E$ with structure group $G_\mathbb{C}$ and the hypermultiplets transform as covariantly constant sections of the associated vector bundle in the representation $T^*M$. The stability condition on $(X, Y)$ from the real moment map equation will require a certain number of linearly independent, covariantly constant, holomorphic sections of $E$. This requirement translates into the existence of a holomorphically trivial subbundle spanned by those sections.

Under Assumption I of section 3 and with a generic FI parameter $\zeta \in \mathfrak{c}_H$, the whole bundle $E$ is trivialised and with it the underlying principal bundle $P$. Then we can choose $d_A = d$, Hitchin's equation becomes trivial, the hypermultiplet fields are constant and the entire system reduces to the equations defining the Higgs branch $X$,

$$
\mu^{AB} = \zeta^{AB}. \tag{157}
$$

In summary, the FI parameter forces the supersymmetric quantum mechanics onto a pure hypermultiplet branch. We focus here on this case only, leaving the a more general description to future work.

Let us now consider massless fluctuations around a point $p \in X$ on the Higgs branch. There are of course 1d $\mathcal{N} = (0, 4)$ hypermultiplet fluctuations transforming in $T_p X$. In addition, there are 1d $\mathcal{N} = (0, 4)$ Fermi multiplet fluctuations, which are found by expanding Yukawa couplings around $p$ to obtain fermion mass terms. This shows that the remaining massless Fermi fluctuations obey the same linearised equations as the hypermultiplets. However, since the Fermi multiplets transform as 1-forms on $\Sigma$, they generate $g$ copies of the tangent space $T_p X$.

It is convenient to introduce a derived moduli space $\mathfrak{M}$ whose tangent complex at a point reproduces the massless fluctuations of both 1d $\mathcal{N} = (0, 4)$ hypermultiplets and Fermi multiplets. Let us write $\mathcal{E} := \mathbb{C}^g \otimes T_X$. Then

$$
\mathfrak{M} = \operatorname{Spec} \operatorname{Sym}^\bullet(\mathcal{E}^\vee[1]) \tag{158}
$$

such that the tangent complex

$$T_{\mathfrak{M}} = T_X \oplus \mathcal{E}[-1] \tag{159}$$

reproduces the correct fluctuations of hypermultiplets and Fermi multiplets, including the correct shift of cohomological degree for the Fermi multiplets. In more standard terminology, the effective supersymmetric quantum mechanics is a smooth $\mathcal{N} = (0,4)$ sigma model with hyper-Kähler target $X$ and hyper-holomorphic vector bundle $\mathcal{E}$.

## 6.3 Supersymmetric Ground States

Let us now consider the supersymmetric ground states. Since we are restricted to the flux-zero sector, supersymmetric ground states are uncharged under the topological symmetry $T_C$ and $\gamma_C = 0$. We then consider the supersymmetric ground states of the $\mathcal{N} = (0,4)$ supersymmetric quantum mechanics with target $X$ and vector bundle $\mathcal{E}$.

The states of the supersymmetric quantum mechanics consist of $L^2$-normalisable sections of

$$\Omega_X^{0,\bullet} \otimes \mathcal{F}, \tag{160}$$

where

$$\mathcal{F} = \sqrt{K_X} \otimes \widehat{\mathrm{Sym}}^{\bullet}(\mathcal{E}[-1]). \tag{161}$$

The supercharges act on sections by

$$Q_+ = \bar{\partial}_{\mathcal{F},I} \qquad Q_- = (\bar{\partial}_{\mathcal{F},-I})^{\dagger}, \tag{162}$$

where $I$ is the default complex structure with holomorphic coordinates $(X, Y)$ and $-I$ is the conjugate complex structure with holomorphic coordinates $(\bar{Y}, -\bar{X})$. The supersymmetric ground states are in principle $L^2$ forms annihilated by all four supercharges or equivalently harmonic forms for the Dolbeault Laplacian twisted by $\mathcal{F}$.

If the target space $X$ were compact, the supersymmetric quantum mechanics would be gapped and supersymmetric ground states could be understood as the cohomology of any one supercharge. In particular, considering the cohomology of the supercharge $Q_+$, we could drop the $L^2$ condition and write

$$\mathcal{H} = H_{\bar{\partial}}^{0,\bullet}(X, \mathcal{F}). \tag{163}$$

However, the Higgs branch $X$ is non-compact and, as it stands, the spectrum of this supersymmetric quantum mechanics is not gapped and a cohomological description is not available. To remedy this situation, we now turn back on the mass parameters $m$.

## 6.4 Mass Parameters

Let us now introduce mass parameters $m$. This effectively shifts the real vectormultiplet scalar $\sigma \to \sigma + m$, such that the supersymmetric quantum mechanics localises to solutions of the same equations (156) except that now

$$(\sigma + m)X^A = (\sigma + m)X^{\dagger A} = 0. \tag{164}$$

This remaining solutions are those invariant $U(1)_m \subset T_H$ action generated by the mass parameter. Under Assumption I, this corresponds to the $U(1)_m$ fixed locus on $X$. Under Assumption II, for generic mass parameter $m \in \mathfrak{c}_H$, this fixed locus is an isolated set of points $p_I$. We then expect the supersymmetric ground states to be obtained by quantising the hypermultiplet and Fermi multiplet flutuations around the points $p_I$.

Let us re-phrase this from the perspective of the finite-dimensional supersymmetric quantum mechanics with hyper-Kähler target space $X$. The real mass corresponds to introducing a perfect Morse-Bott function

$$h = m \cdot \mu_{\mathbb{R},H}, \tag{165}$$

which is the real moment map for action of $U(1)_m$ on $X$. This has the effect of conjugating the supercharges in the supersymmetric sigma model such that

$$Q_+ = e^{-h} \bar{\partial}_{\mathcal{F},I} e^h := \bar{\partial}_m, \qquad Q_- = e^{-h} \bar{\partial}_{\mathcal{F},-I}^\dagger e^h. \tag{166}$$

The supersymmetric ground states are now tri-harmonic forms for the mass-formed Dolbeault Laplacians. Provided the fixed locus $X^{U(1)_m}$ is compact, the spectrum of the supersymmetric quantum mechanics is gapped and supersymmetric ground states can be identified with the cohomology of $Q_+$, namely

$$\mathcal{H} = H_{L^2, \bar{\partial}_m}^{0,\bullet}(X, \mathcal{F}). \tag{167}$$

Although this result is reasonable, it is not suitable for computation.

A useful computational approach involves sending the real mass parameters $m$ to infinity in a given chamber. Intuitively, in this limit the wavefunctions are localised around the fixed points $p_I$, leading to a Fock space of exact perturbative ground states attached to each $p_I$ by quantizing a massive sigma model with target $T_{p_I} X$ and trivial hyperholomorphic bundle $\mathcal{F}_p$. This should then be supplemented by holomorphic instanton corrections. This picture originates from [33] and can be formulated algebraically as a Cousin complex for the stratification of $X$ induced by the $\mathbb{C}_m^*$ action [14–17].

We will argue, however, that instanton corrections are absent in models with $\mathcal{N} = (0,4)$ supersymmetry and therefore the supersymmetric ground states are fully captured by Fock spaces attached to each $p_I$. The result for a general theory satisfying the assumptions of section 3 can therefore be reduced to a computation involving free hypermultiplets parametrising $T_{p_I} X$.

For this reason, in the following section we first consider a single free hypermultiplet, SQED with $N$ hypermutiplets, and finally the general class of theories satisfying the assumptions of section 3.

# 7 B-Twist Examples

## 7.1 Hypermultiplet

For a free hypermultiplet, the effective supersymmetric quantum mechanics contains the following supermultiplets:

- A $\mathcal{N} = (0,4)$ hypermultiplet $\Phi^a$, from $H^0(\mathcal{O})$.

- $g$ $\mathcal{N} = (0,4)$ Fermi multiplets $\chi_i^a$, from $H^1(\mathcal{O})$, $i \in \{1, \dots, g\}$

both transforming as doublets of $G_H \cong SU(2)$ global symmetry. This is a free supersymmetric quantum mechanics with target $X = T^*\mathbb{C}$ and $\mathcal{E} = \mathbb{C}^g \otimes T_X$.

Let us determine the weights of these fluctuations. First, the cohomological grading $F$ a priori corresponds to $R_H$. However, for reasons discussed in section 2.2, we define a new R-symmetry

$$\widetilde{R}_H = R_H - J_H, \tag{168}$$

where $J_H$ is the generator of the flavour symmetry $T_H \cong U(1)$ and define $F := \widetilde{R}_H$. This means that the bosonic components of the hypermultiplet $\phi^1 = X$, $\phi^2 = Y$ have weights

$F = 0, 2$, while the $g$ Fermi multiplet components $\chi_i^1$, $\chi_i^2$ have weights $F = -1, +1$. This is now compatible with $(-1)^F$ as the fermion number. The secondary grading is then similarly $R_+ = \frac{1}{2}(\widetilde{R}_H - R_C)$. The weights of top components of the supermultiplets are summarised in table.

Table 3: Weights of hypermultiplet fields in the $B$-twist.

|       | $\phi^1$ | $\phi^2$ | $\chi^1$ | $\chi^2$ |
|-------|----------|----------|----------|----------|
| $F$   | 0        | +2       | −1       | +1       |
| $R_+$ | 0        | +1       | 0        | +1       |
| $J_H$ | +1       | −1       | +1       | −1       |

Since this supersymmetric quantum mechanics consists of a free hypermultiplet and Fermi multiplets, it is possible to determine the supersymmetric ground state wavefunctions exactly by demanding they are annihilated by all four supercharges. The normalisable ground state wavefunctions for the hypermultiplet are

$$
\begin{aligned}
X^{k_1} \bar{Y}^{k_2} e^{-m(|X|^2+|Y|^2)} d\bar{Y}, && m > 0, \\
\bar{X}^{k_1} Y^{k_2} e^{m(|X|^2+|Y|^2)} d\bar{X}, && m < 0,
\end{aligned}
\tag{169}
$$

or equivalently

$$
\begin{aligned}
\left(\frac{\partial}{\partial \bar{X}}\right)^{k_1} \left(\frac{\partial}{\partial Y}\right)^{k_2} e^{-m(|X|^2+|Y|^2)} d\bar{Y}, && m > 0, \\
\left(\frac{\partial}{\partial X}\right)^{k_1} \left(\frac{\partial}{\partial \bar{Y}}\right)^{k_2} e^{m(|X|^2+|Y|^2)} d\bar{X}, && m < 0,
\end{aligned}
\tag{170}
$$

with arbitrary integers $k_1, k_2 \geq 0$. These states are supplemented by wavefunctions for the $g$ Fermi multiplets, which for the $i$-th Fermi multiplet take the form

$$
\begin{aligned}
1, \chi_i^1, \bar{\chi}_i^2, \chi_i^1 \bar{\chi}_i^2, && m > 0, \\
1, \chi_i^2, \bar{\chi}_i^1, \chi_i^2 \bar{\chi}_i^1, && m < 0,
\end{aligned}
\tag{171}
$$

where the choice of Fock vacuum is determined by the sign of the mass of each fermion and omitted from the notation.

Combining the possible wavefunctions from the hypermultiplet and Fermi multiplets and taking into account the weights in table 3, we find

$$
\mathcal{H} = \begin{cases} \widehat{\mathrm{Sym}}^\bullet V & m > 0 \\ \widehat{\mathrm{Sym}}^\bullet V^\vee & m < 0 \end{cases},
\tag{172}
$$

where

$$
V = x(\mathbb{C} \oplus t^{-1}\mathbb{C}^g[1] \oplus \mathbb{C}^g[1] \oplus t^{-1}\mathbb{C}[2]).
\tag{173}
$$

The space of supersymmetric ground states is a Fock space generated by descendants of $X$ (when $m > 0$) and $Y$ (when $m < 0$) integrating over homology classes of $\Sigma$.

Computing the trace over supersymmetric ground states in either chamber, we find

$$
\mathcal{I}^B = (-t^{-\frac{1}{2}} x)^{1-g} (1-x)^{g-1} (1-t^{-1}x)^{g-1},
\tag{174}
$$

which coincides with the $B$-twisted index of a free hypermultiplet with our modified R-charge assignments [7]. In this instance, $\mathcal{H} = \mathcal{H}_{1/2}$ and the general index only receives contributions from supersymmetric ground states.

## 7.2 SQED, 1 hypermultiplet

Now consider $G = U(1)$ with one hypermultiplet of charge $+1$ and topological symmetry $T_C \cong U(1)$. Introducing a 1d FI parameter $\zeta \neq 0$, the supersymmetric quantum mechanics localises to constant configurations for the hypermultiplet fields satisfying

$$|X|^2 - |Y|^2 = \zeta \qquad XY = 0. \tag{175}$$

There is a single solution given by

$$
\begin{aligned}
\zeta > 0 &: \quad X = \sqrt{+\zeta} \qquad Y = 0, \\
\zeta < 0 &: \quad Y = \sqrt{-\zeta} \qquad X = 0,
\end{aligned} \tag{176}
$$

and therefore we expect a single supersymmetric ground state, $\mathcal{H} = \mathbb{C}$.

The generic $B$-twisted index is

$$\mathcal{I} = -(t^{-\frac{1}{2}} - t^{\frac{1}{2}})^{1-g} \sum_{d \in \mathbb{Z}} \xi^d \int_\Gamma \frac{dz}{2\pi i z} \left( \frac{z - t^{\frac{1}{2}}}{1 - t^{\frac{1}{2}} z} \right)^d \left( \frac{t^{\frac{1}{2}} z}{(1 - t^{\frac{1}{2}} z)(z - t^{\frac{1}{2}})} \right)^{1-g} H^g, \tag{177}$$

where

$$H = \frac{(1-t)z}{(z - t^{\frac{1}{2}})(1 - t^{\frac{1}{2}} z)} \tag{178}$$

and evaluates to 1.

## 7.3 SQED, $N$ hypermultiplets

Now consider $G = U(1)$ with $N$ hypermultiplets of charge $+1$. The flavour symmetries are $T_H \cong U(1)^N / U(1)$ and $T_C \cong U(1)$. Introducing a 1d FI parameter in the chamber $\mathfrak{c}_H = \{\zeta > 0\}$, the supersymmetric quantum mechanics localises to constant configurations for the hypermultiplet fields satisfying

$$\sum_{j=1}^N \left( |X_j|^2 - |Y_j|^2 \right) = \zeta, \qquad \sum_{j=1}^N X_j Y_j = 0. \tag{179}$$

We therefore have an $\mathcal{N} = (0,4)$ supersymmetric quantum mechanics with target space $X = T^* \mathbb{CP}^{N-1}$ and as usual $\mathcal{E} = \mathbb{C}^g \otimes T_X$.

As $X$ is non-compact, this supersymmetric quantum mechanics is not gapped. We introduce mass parameters $(m_1, \ldots, m_N)$ for the flavour symmetry. For generic values of the mass parameters, there are $N$ isolated fixed points, which are labelled by a choice of hypermultiplet $I = \{i\}$.

Concretely, the fluctuations around a fixed point $I$ consist of

- $(N-1)$ $\mathcal{N} = (0,4)$ hypermultiplets of $T_H$ weights $x_j x_i^{-1}$, $j \neq i$.

- $g \times (N-1)$ $\mathcal{N} = (0,4)$ Fermi multiplets of $T_H$ weights $x_j x_i^{-1}$, $j \neq i$.

We can therefore recycle the results from a free hypermultiplet to compute the space of supersymmetric ground states. Let us select the chamber $\mathfrak{c}_H = \{m_1 > m_2 > \cdots m_N\}$ such that $m_j - m_i > 0$ when $j > i$. At a fixed point $I$, the tangent bundle decomposes as

$$T_{p_I} X = N_I^+ \oplus N_I^-, \tag{180}$$

where

$$N_I^+ := \bigoplus_{j>i} \frac{x_i}{x_j}\mathbb{C} \oplus \bigoplus_{j<i} t^{-1}\frac{x_j}{x_i}\mathbb{C}[2],$$
$$N_I^- := \bigoplus_{j<i} \frac{x_i}{x_j}\mathbb{C} \oplus \bigoplus_{j>i} t^{-1}\frac{x_j}{x_i}\mathbb{C}[2],$$

(181)

encode positive and negative weights for $U(1)_m \subset T_H$. Here for convenience we have also made the other gradings manifest.

In view of the result (172) for a free hypermultiplet, the perturbative supersymmetric ground states arising from each fixed point are

$$\mathcal{H}_I = \widehat{\mathrm{Sym}}^\bullet V_I,$$

(182)

where

$$V_I := N_I^+ \oplus \left(N_I^-\right)^\vee \oplus \left(N_I^+[-1] \oplus \left(N_I^-[-1]\right)^\vee\right) \otimes \mathbb{C}^g.$$

(183)

Since states at different fixed points have different flavour weights, we conclude that there are no instanton corrections and the space of true supersymmetric ground states is a direct sum of contributions from each fixed point,

$$\mathcal{H} = \bigoplus_{I=1}^N \mathcal{H}_I.$$

(184)

We now check compatibility with limits of the twisted index. The general $B$-twisted index with our charge assignements is

$$\mathcal{I} = -(t^{-\frac{1}{2}}-t^{\frac{1}{2}})^{1-g}\sum_{d\in\mathbb{Z}}\xi^d \int_\Gamma \frac{dz}{2\pi i z} \prod_{j=1}^N \left(\frac{zx_j - t^{\frac{1}{2}}}{1-t^{\frac{1}{2}}zx_j}\right)^d \left(\frac{t^{\frac{1}{2}}zx_j}{(1-t^{\frac{1}{2}}zx_j)(zx_j - t^{\frac{1}{2}})}\right)^{1-g} H^g,$$

(185)

where $H$ is the Hessian determinant. For $\zeta > 0$, the contour $\Gamma$ selects the poles at $z = t^{-\frac{1}{2}}x_j^{-1}$ and the index only receives contributions from $d > g-1$. Therefore, in the limit $\xi \to 0$, the index only receives a contribution from zero flux, $d = 0$. The result is

$$\lim_{\xi\to 0}\mathcal{I} = \sum_{i=1}^N \prod_{j\neq i}\left(-t^{-\frac{1}{2}}\frac{x_i}{x_j}\right)^{1-g}\left(1-\frac{x_i}{x_j}\right)^{g-1}\left(1-t^{-1}\frac{x_i}{x_j}\right)^{g-1}.$$

(186)

By the general mechanism described in section 2.5, this contribution is captured by supersymmetric ground states. Indeed, this result is in perfect agreement with the graded trace over supersymmetric ground states found above. We note that there are no cancelations between contributions from each pole $z = t^{-\frac{1}{2}}x_j^{-1}$, which reflects the absence of instanton corrections to supersymmetric ground states.

## 7.4 General Class

Let us now consider the general class of theories satisfying the assumptions of section 3. As always we select a pair of chambers $(\mathfrak{c}_H, \mathfrak{c}_C)$. The FI parameter $\zeta \in \mathfrak{c}_C$ determines a Higgs branch $X$. We assume a reasonable definition of cohomological grading $F$ such that the holomorphic symplectic form on $X$ has cohomological degree $+2$ and secondary degree $+1$. The mass parameters $m \in \mathfrak{c}_H$ determines a $U(1)_m$ isometry with isolated fixed points labelled by an index $I$.

In similarity to the previous example, let us decompose the tangent space at a fixed point into positive and negative weight spaces for $U(1)_m$

$$T_{p_I} X = N_I^+ \oplus N_I^- \,, \tag{187}$$

where

$$(N_I^+)^\vee = t N_I^- [-2] \tag{188}$$

due the weights of the holomorphic symplectic form on $X$. By considering the fluctuations around each fixed point, the perturbative supersymmetric ground states are

$$\mathcal{H} = \bigoplus_I \widehat{\mathrm{Sym}}^\bullet \left[ N_I^+ \oplus (N_I^-)^\vee \oplus \left( N_I^+[-1] \oplus (N_I^-[-1])^\vee \right) \otimes \mathbb{C}^g \right] . \tag{189}$$

As a sanity check, let us check that this reproduces the result for a hypermultiplet. In the chamber $\mathfrak{c}_H = \{ m > 0 \}$, we have

$$
\begin{aligned}
N_I^+ &= \mathbb{C} \langle \frac{\partial}{\partial Y} \rangle = x t^{-1} \mathbb{C}[2], \\
N_I^- &= \mathbb{C} \langle \frac{\partial}{\partial X} \rangle = x^{-1} \mathbb{C}
\end{aligned}
\tag{190}
$$

and therefore

$$\mathcal{H} = \widehat{\mathrm{Sym}}^\bullet \left[ x \left( \mathbb{C} \oplus t^{-1} \mathbb{C}[2] \oplus \mathbb{C}^g[1] \oplus t^{-1} \mathbb{C}^g[1] \right) \right] , \tag{191}$$

which is consistent with section 7.1.

It remains to be argued that there are no instanton corrections between states associated to different fixed points. Instanton corrections can occur only between states that share the same flavour weight. If at a fixed point this weight is positive, at the other it must be negative. Due to (188), it is impossible for the cohomological supercharge $Q_+$ to relate such states. We therefore conclude that instanton corrections must be absent.

Finally, from the supersymmetric ground states (189) we can immediately compute limits of the twisted index, which we do here for simplicity in the limit $t \to 1$. Let $\Phi_I^+$ denote the set of positive weights at the fixed point $I$. Then

$$\lim_{t \to 1} \mathcal{I} = \mathrm{Tr}_{\mathcal{H}} (-1)^F x^{J_H} = \sum_I \prod_{\lambda \in \Phi_I^+} (-1)^{(1-g)d_I} \left( \frac{x^{\lambda/2}}{1 - x^\lambda} \right)^{2-2g} , \tag{192}$$

where $d_I$ can be obtained by considering the $(-1)^F$ grading of the tangent space.

## 8 Mirror Symmetry

3d $\mathcal{N} = 4$ gauge theories enjoy an infrared duality called mirror symmetry [34]. It relates pairs of theories $(\mathcal{T}, \mathcal{T}^\vee)$ that flow to the same superconformal fixed point in the infrared. At the level of the supersymmetry algebra, the duality acts as an involution that exchanges the R-symmetries $SU(2)_H$ with $SU(2)_C$ and the flavour symmetries $T_H$ and $T_C$. Mirror symmetry implies non-trivial relationships between mathematical structures associated to pairs of 3d $\mathcal{N} = 4$ theories [35–38].

In the context of this paper, mirror symmetry exchanges the $A$ twist of a theory $\mathcal{T}$ with the $B$ twist of its mirror $\mathcal{T}^\vee$. In particular, it exchanges the parameters appearing in the twisted index according to $(t, y, \xi) \leftrightarrow (t, \xi, y)$. Mirror symmetry for the twisted index reads

$$\mathcal{I}^A[\mathcal{T}](t, y, \xi) = \mathcal{I}^B[\mathcal{T}^\vee](t, \xi, y) \,. \tag{193}$$

In the conventions of section 3, the R-symmetry conjugate to $t$ is defined relative to the twist, namely $R = \frac{1}{2}(R_C - R_H)$ in the $A$-twist and $R = \frac{1}{2}(R_H - R_C)$ in the $B$-twist. This the parameter $t$ transforms to itself under mirror symmetry in contrast to [7].

This lifts to a statement about mirror symmetry for supersymmetric ground states,

$$\mathcal{H}^A[\mathcal{T}] = \mathcal{H}^B[\mathcal{T}^\vee], \tag{194}$$

where the double R-symmetry grading is exchanged according to $(R_H, R_C) \leftrightarrow (R_C, R_H)$, or equivalently $(F, R) \leftrightarrow (F, R)$, the flavour gradings are exchanged $T_C \leftrightarrow T_H$ and finally the chambers are exchanged according to $(\mathfrak{c}_C, \mathfrak{c}_H) \leftrightarrow (\mathfrak{c}_C, \mathfrak{c}_H)$.

The computations of supersymmetric ground states in this paper involve strikingly different mathematical formulations in the $A$ and $B$ twist. Thus mirror symmetry makes non-trivial predictions of equalities between these constructions.

From the above examples, it can readily be checked that upon fixing chambers $\mathfrak{c}_c, \mathfrak{c}_c$ the most basic mirror symmetry, relating SQED[1] and the free hypermultiplet, holds at the level of the Hilbert spaces (as graded vector spaces). The same holds for the self-mirror property SQED[2]: the A-twist Hilbert space is isomorphic to the B-twist Hilbert space. More generally, since mirror symmetry implies that the number of fixed points of the Higgs branches of two mirror theories is the same (they correspond to the same vacua), it is tempting to think the symmetry pairs the Hilbert spaces associated to fixed points. The formal similarity of the Hilbert spaces associated to fixed points in the A- and B- twist (see (151) and (189)) further suggests that this is the case. We show this explicitly for abelian theories at the level of flavour and topological grading in appendix B, and leave a more general discussion to future work.

## 9 Discussion

In this section, we discuss connections to other work and directions for future investigation.

- An important next step is to introduce background expectation values for vectormultiplet and twisted vectormultiplets for the flavour symmetries $T_H$ and $T_C$ respectively along the Riemann surface $\Sigma$. The task is then to understand and compute the associated supersymmetric Berry connections for the bundle of supersymmetric ground states in the $\mathcal{N} = 4$ supersymmetric quantum mechanics.

  In the $A$-twist, one can introduce a holomorphic bundle for $T_H$ and a complex flat connection for $T_C$. On the other hand in the $B$-twist, one can introduce a holomorphic bundle for $T_C$ and a complex flat connection for $T_H$. The structure of the supersymmetric Berry connections should follow the constructions of [22, 24].

- The computation of supersymmetric ground states can be enriched by adding line operators at points $p \in \Sigma$ and preserving the same 1d $\mathcal{N} = 4$ supersymmetry algebra as the $A$-twist or the $B$-twist. This class of line operators have been studied in [39, 40].

- Coulomb and Higgs branch local operators and their descendents wrapped on cycles in $\Sigma$ lead to operators in the effective supersymmetric quantum mechanics that act on supersymmetric ground states. In future work, we will show that the supersymmetric ground states $\mathcal{H}_A$, $\mathcal{H}_B$ transform as modules for the factorisation homology on $\Sigma$ of the Coulomb and Higgs branch chiral rings respectively.

- It would be interesting to connect the construction of supersymmetric ground states $\mathcal{H}_A$, $\mathcal{H}_B$ here to spaces of conformal blocks for the vertex algebras $\mathcal{V}_A$, $\mathcal{V}_B$ studied in [22, 41, 42]. A key difference is the need in this paper to introduce real FI and mass parameters

that break the $SU(2)_H$ and $SU(2)_C$ R-symmetries needed to perform the full topological $A$-twist and $B$-twist respectively. The breaking is compatible with the 3d $\mathcal{N} = 2$ topological-holomorphic twist discussed in [43].

- Finally, related to the previous points, it would be interesting to understand the space of supersymmetric ground states in the context of boundary conditions for $\mathcal{N} = 4$ SYM investigate potential connections to geometric Langlands [44–46]. If a 3d $\mathcal{N} = 4$ supersymmetric gauge theory can be realised by compactification on an interval $I$ with half-BPS boundary conditions $B_L, B_R$, the spaces of supersymmetric ground states should correspond to morphisms between objects associated to $B_L, B_R$ in the category of boundary conditions in the Kapustin-Witten twist on $\Sigma$. Again, our setup departs slightly from this picture due to the need to introduce boundary FI and mass parameters.

## Acknowledgments

We gratefully acknowledge discussions with Stefano Cremonesi, Tudor Dimofte. The work of MB is supported by the EPSRC Early Career Fellowship EP/T004746/1 "Supersymmetric Gauge Theory and Enumerative Geometry" and the STFC Research Grant ST/T000708/1 "Particles, Fields and Spacetime".

## A  Cohomology of Symmetric Products

The cohomology of the symmetric product was studied in ref. [31]. It has $2g$ fermionic generators $\xi^i$ and $\tilde{\xi}^i$, $i \in \{1, \dots g\}$ of bidegree $(1, 0)$ and $(0, 1)$ respectively, as well as a bosonic generator of bidegree $(1, 1)$. As a graded vector space, we have for $p + q \leq n$

$$\mu^p \nu^q H^{p,q}\left(\mathrm{Sym}^d(\Sigma)\right) \cong \bigoplus_{i=0}^{\min(p,q)} S^i(\mu\nu\mathbb{C}) \otimes \wedge^{p-i}(\mu\mathbb{C}^g) \wedge \wedge^{q-i}(\nu\mathbb{C}^g), \tag{195}$$

where $\mu$, $\nu$ are grading parameters. It follows that

$$\sum_{d \in \mathbb{N}} x^d \left( \mu\nu \sum_{p,q} H^{p,q}\left(\mathrm{Sym}^d(\Sigma)\right) \right) \cong S^\bullet(x\mathbb{C} \oplus \mu\nu x\mathbb{C}) \otimes \wedge^\bullet(\mu x\mathbb{C}^g) \wedge \wedge^\bullet(\nu x\mathbb{C}^g) \tag{196}$$

for another grading paramter $x$. By taking a graded trace, we get that $h^{p,q}\left(\mathrm{Sym}^d(\Sigma)\right)$ is the coefficient of $x^d \mu^p \nu^q$ in the series expansion of

$$\frac{(1 + \mu x)^g (1 + \nu x)^g}{(1 - x)(1 - \mu\nu x)} \tag{197}$$

around $x = 0$. Restricting to the grading by the fermion number, which amounts to setting $\mu = \nu = -1$, we can derive a generating function for the Euler-Poincaré characteristic

$$\sum_{d \in \mathbb{N}} x^d \left( \sum_{k=0}^d (-1)^k H_{dR}^k\left(\mathrm{Sym}^d(\Sigma)\right) \right) = (1 - x)^{2(g-1)}. \tag{198}$$

## B  Abelian Mirror Symmetry

Let us consider abelian theories subject to the assumptions spelled out in (3.1). The number of hypermulitplets will be denoted by $N$, the rank by $k$. In order to be consistent with our

previous notation, we will denote the charges of their $(X, Y)$ components by $(\rho, -\rho)$. Notice that since we are considering quiver gauge theories, all the non-vanishing entries in $\rho$ are $\pm 1$. We will also denote by $(\lambda, -\lambda)$ the charges under a maximal torus of the flavour symmetry $T_H$, which has rank $N - k$. Together with the gauge charges, these form a $N \times N$ charge matrix

$$\mathbf{Q} = \begin{pmatrix} \rho^T \\ \lambda^T \end{pmatrix}. \tag{199}$$

The flavour weights $\lambda$ are only defined up to the gauge weights $\rho$, and we can set $\det(\mathbf{Q}) = 1$.

The relation between charge matrices of two mirror dual abelian theories $\mathcal{T}$ and $\tilde{\mathcal{T}}$ is known [47]:

$$\begin{pmatrix} \widetilde{\lambda}^T \\ \widetilde{\rho}^T \end{pmatrix} = \begin{pmatrix} \rho^T \\ \lambda^T \end{pmatrix}^{-1,T}. \tag{200}$$

We can use this to prove that mirror symmetry identifies the A-twist Hilbert space of a theory $\mathcal{T}$ to the the B-twist Hilbert space of a theory $\tilde{\mathcal{T}}$, and vice-versa. Moreover, the isomorphism maps the contribution from a fixed point to the contribution of a mirror dual fixed point.

Recall that to a fixed point $I$ we associate in particular a selections of $k$ hypermultiplets such that the submatrix $\rho_I$ of $\mathbf{Q}$ is non-singular. We can split the mirror symmetry relation

$$\left( \begin{array}{c|c} \rho_I^T & \rho_{I^\vee}^T \\ \hline \lambda_I^T & \lambda_{I^\vee}^T \end{array} \right) \left( \begin{array}{c|c} \widetilde{\lambda}_I & \widetilde{\rho}_I \\ \hline \widetilde{\lambda}_{I^\vee} & \widetilde{\rho}_{I^\vee} \end{array} \right) = \mathbf{1}_{N,N}. \tag{201}$$

Multiplication of the blocks yields

$$\begin{aligned}
\rho_I^T \widetilde{\lambda}_I + \rho_{I^\vee}^T \widetilde{\lambda}_{I^\vee} &= \mathbf{1}_{k,k}, \\
\rho_I^T \widetilde{\rho}_I + \rho_{I^\vee}^T \widetilde{\rho}_{I^\vee} &= 0, \\
\lambda_I^T \widetilde{\rho}_I + \lambda_{I^\vee}^T \widetilde{\rho}_{I^\vee} &= \mathbf{1}_{N-k,N-k}, \\
\lambda_I^T \widetilde{\lambda}_I + \lambda_{I^\vee}^T \widetilde{\lambda}_{I^\vee} &= 0.
\end{aligned} \tag{202}$$

From this we can conclude

$$\begin{aligned}
\widetilde{\rho}_{I^\vee}^T \left( -\rho_{I^\vee} \rho_I^{-1} \lambda_I + \lambda_{I^\vee} \right) &= \left( \widetilde{\rho}_I^T \rho_I \right) \left( \rho_I^{-1} \lambda_I \right) + \widetilde{\rho}_{I^\vee}^T \lambda_{I^\vee} \\
&= \widetilde{\rho}_I^T \lambda_I + \mathbf{1}_{N-k,N-k} - \widetilde{\rho}_I^T \lambda_I \\
&= \mathbf{1}_{N-k,N-k},
\end{aligned} \tag{203}$$

where we used the second and third of the equations in (202). Thus

$$\widetilde{\rho}_{I^\vee}^{T,-1} = -\rho_{I^\vee} \rho_I^{-1} \lambda_I + \lambda_{I^\vee}. \tag{204}$$

The rows of the RHS of this expression correspond precisely to the tangent flavour weights of theory $\mathcal{T}$ at fixed point $I$. Since the gauge weights in $\rho_I$ must satisfy the JK condition (51), it follows that according to the mirror chamber for the masses the flavour weights are positive.

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
