# Peer review of "Supersymmetric Ground States of 3d $\mathcal{N}=4$ Gauge Theories on a Riemann Surface"

_SciPost Physics, doi:SciPost Phys. 12, 072 (2022)_

## Round 2 · Referee Report · Anonymous (Referee 1) · 2021-10-28

Report

I think the paper easily meets the acceptance criteria for this journal and
I recommend its publication. However, I have a few questions/comments (listed in the report) that I would like the authors to address, prior to publication.

Attachment

  • validity: -
  • significance: -
  • originality: -
  • clarity: -
  • formatting: -
  • grammar: -

Author:  Heeyeon Kim  on 2021-11-19  [id 1961]

(in reply to Report 1 on 2021-10-28)

The authors would like to thank the referee for the comments and suggestions.

  1. In eq. (5.9), we consider a non-trivial background flux for the $U(1)$ flavor symmetry that corresponds to the line bundle $L$ with degree $d$ in eq. (5.1). On the other hand, for simplicity, we turn off the flux for the $U(1)$ topological symmetry in section 7.2. Mirror symmetry would work for all $d$ once we consider non-trivial backgrounds for the topological symmetry in the B-twist.

  2. The large mass limit is indeed a computational trick in the B-twist and is not related to the large $\widetilde \zeta$ limit in the A-twist. The mass parameter is dual to $\zeta$, which is held fixed in a given chamber. Taking a large mass limit does not change the result as long as we stay in a chamber in the space of real mass parameters.

  3. We expect that the mirror symmetry holds in the limit $e^2\rightarrow \infty, \zeta\rightarrow 0$. However, in this limit, non-compact directions emerge in the field space, which renders the space of supersymmetric ground states ill-defined.

  4. Reference added on page 20, typos corrected.

---

## Round 2 · Referee Report · Anonymous (Referee 2) · 2021-11-1

Report

The study of 3d $N=4$ theories has been one of the recent focal points of the interaction between physics and mathematics. In the manuscript, the authors study spaces of supersymmetric ground states of certain class of 3d $N=4$ gauge theories on a Riemann surface, with partial A- or B-twist, and give them precise geometric interpretations. The result is interesting and novel, and I find the exposition systematic and detailed with many worked out examples. I believe the manuscript has surpassed the expectation and criteria for this journal and recommend it for publication.

I would like to entertain below a few ideas for further improving the manuscript:

-- I find the Introduction to be a good summary of the content of the paper. However, as the assumptions on the Coulomb and the Higgs branches in the A- and B-twist cases respectively are important parts of the scientific results of this work, maybe it a good idea to not ``bury the lead'' but instead make these assumptions more visible by including them in the Introduction.

-- It might worth clarifying what the precise meaning of ``finite-dimensional $N = 4$ supersymmetric quantum mechanics'' is in the second paragraph of Section 1.1. With finite-dimensional target, the full Hilbert space of the SQM can be still infinite dimensional.

-- On the same page, the condition that $d$ is larger than $g$ seems only make sense for $U(1)$ subgroups of $G$.

-- The authors find the Berry connection in the space of parameters is flat. Since this space can be non-simply-connected due to values of parameters that close the gap in the SQM spectrum, there is still the possibility of having non-trivial monodromies. It might be interesting for the author to comment on whether (or when) they exist.

-- The mirror symmetry at the level of SQM ground states is a very interesting subject. However, the only example given in Section 8 is self-mirror. It might be a good idea to include another example with a more non-trivial mirror pair.
  • validity: -
  • significance: -
  • originality: -
  • clarity: -
  • formatting: -
  • grammar: -

Author:  Heeyeon Kim  on 2021-11-19  [id 1962]

(in reply to Report 2 on 2021-11-01)
Category:
answer to question
correction

The authors would like to thank the referee for the comments and suggestions.

  1. We added a paragraph on the first page stating our assumptions.

  2. In response to comments 2 and 3, we changed the corresponding sentences in the second paragraph and in the beginning of the fourth paragraph in section 1.1.

  3. Singularities are real co-dimension one hyperplanes in the space of real mass parameters and therefore there's no notion of monodromy for the Berry connections.

  4. The mirror symmetry is shown for the general abelian quivers in appendix B. It is straightforward to combine the analysis of SQCD discussed in detail in the main body of the paper to understand more non-trivial mirror examples.

---

## Round 3 · List of Changes

1. On page 1, we added a paragraph after bullet points, which explicitly states our assumptions.

  2. In section 1.1, we corrected the second sentence of the second paragraph, from "...finite-dimensional..." to "...finite-dimensional and compact...".

  3. In the same section, we corrected the first sentence of the fourth paragraph, from "when the degree d is large.." to "when all the components of the degree d is large.."

  4. Reference [28] added on page 20

  5. At the bottom of page 37, we changed \times to \otimes

  6. Above (A.1), we added "for p+q<= n".

---

## Editorial Decision

published